



# Influence of the spatial distribution of gravity wave activity on the middle atmospheric circulation and transport.

Petr Šácha[1], Friederike Lilienthal[2], Christoph Jacobi[2], Petr Pišoft[1]

[1]Department of Atmospheric Physics, Faculty of Mathematics and Physics, Charles University in Prague, V Holesovickach 2, Prague 180 00, Czech Republic

[2]University of Leipzig, Institute of Meteorology, Stephanstr. 3, 04103 Leipzig, Germany

*Correspondence to*: Petr Šácha(petr.sacha@mff.cuni.cz)

**Abstract.** Analyzing GPS radio occultation density profiles, we have recently pointed out a localized area of enhanced gravity wave (GW) activity and breaking in the lower stratosphere of the Eastern Asia/North-western Pacific (EA/NP) region. With a mechanistic model for the middle and upper atmosphere (MUAM), experiments are performed to study a possible effect of such a localized IGW breaking region on the large-scale circulation and transport and also more generally, possible influence of spatial distribution of gravity wave activity on the middle atmospheric circulation and transport.

The results indicate an important role of the spatial distribution of GW activity for the polar vortex stability, formation of planetary waves (PW) and for the strength and structure of the zonal mean residual circulation. Also, a possible effect of a zonally asymmetric GW breaking in the longitudinal variability of Brewer-Dobson circulation is analyzed. Finally, consequences of our results for a variety of research topics (Sudden Stratospheric Warmings, atmospheric blocking, teleconnections and a compensation mechanism between resolved and unresolved drag) are discussed.

## 1 Introduction:

Consideration of gravity wave (GW) related processes is necessary for a proper description and modeling of the middle (as reviewed comprehensively by Fritts and Alexander, 2003) and upper atmospheric dynamics (see, e.g., the review by Smith, 2012). In an early numerical attempt to demonstrate the importance of IGWs for the middle atmospheric dynamics, Holton (1983) concluded with formulating some requirements on GW related information (e.g. spatial and seasonal distributions of phase speeds, amplitudes etc.) to improve global climate model parameterization schemes. However, only recently satellite and other global datasets with improved resolution and novel analysis methods together with high-resolution global models have been tightening the constraints for the parameterizations that can improve the treatment of these waves in climate models (Alexander et al., 2010; Geller et al., 2013).

Complex understanding and unbiased modeling of the middle atmospheric conditions is vital for climate research and there is strong evidence that coupling between chemistry and dynamics in the stratosphere is essential for surface climate variability and climate change in both hemispheres (Calvo et al., 2015). Long-term and decadal changes of the middle atmosphere have been found to be largely owing to climate gas changes and thus reflect climate change (e.g. Lübken et al., 2013; Jacobi et al.,



2015) yet it has been difficult to match observations with the robust model stratospheric change projections determined to a large extent by a speeding-up of the Brewer–Dobson circulation (BDC; Butchart, 2014; Kidston et al., 2015). Although weather systems are not able to penetrate into the stratosphere there is a wide recognition of dynamical links between the stratosphere and troposphere
with a potential to significantly affect conditions at the surface (Haynes, 2005; Kidston et al., 2015). Better representation of the stratosphere could improve the long-range (Hardiman and Haynes, 2008) and also short range forecast skills (Gerber et al., 2012).

Using the transformed Eulerian mean equations Dunkerton (1978) derived a first dynamically consistent picture of the mean-transport streamlines for the middle atmosphere that is now generally
considered as the basic concept of the BDC. BDC is still traditionally formulated using the zonal mean two-dimensional framework. However, the longitudinally variable tracer distributions such as $O_3$ and $H_2O$ strongly suggest the need for three-dimensional transport analysis, e.g. strong correlations between the ozone amount and vertical motions of air in the lower stratosphere (Iwao and Hirooka, 2006). Demirhan Bari et al. (2013) found a 3D structure of the circulation in the middle atmosphere to
be in good correspondence with tracer fields, especially in relation to the zonal wave-one pattern observed in the stratosphere and mesosphere. However, the study did not give a comprehensive dynamical explanation of the discovered circulation structures (enhanced downward branch of BDC over north-eastern Asia, wave-one pattern).

PWs are usually thought to be created in the troposphere and then vertically propagating into the
middle atmosphere. The theoretical possibility of PW creation by zonally asymmetric IGW breaking was first numerically analyzed by Holton (1984) and then, e.g., by Smith (2003) and Oberheide et al. (2006), and experimentally verified by Lieberman et al. (2013). There is an agreement in the literature on the role of wave activity in preconditioning the sudden stratospheric warmings (SSWs; e.g. Ayarzaguena et al., 2011).

SSW events belong to the most pronounced atmospheric phenomena with abrupt changes of middle atmospheric circulation and tracer distribution, and they have effects on tropospheric weather patterns (e.g. Manney et al., 2009; Kuroda, 2008; Lehtonen and Karpechko, 2016). SSW dynamics and their impacts differ whether a split or a displacement of the stratospheric polar vortex takes place (Seviour et al. 2016.) and it has been observed that displacements are connected with a dominating wave-one
activity while vortex splits come along with stronger wave-two activity (e.g., Kuttippurath and Nikulin, 2012). Generally, most attention is paid to the role of upward propagating PWs in preconditioning the SSW (Hoffmann et al., 2007; Nishii et al., 2009; Alexander and Shepherd, 2010).

The two open questions regarding the dynamics of SSWs are: what types of wave phenomena are responsible for the SSW triggering and what are the necessary basic state conditions? There are two
main triggering theories discussed - anomalous tropospheric upward wave fluxes or nonlinear resonance in connection to the vortex geometry (Albers and Birner, 2014). Also, there is growing observational evidence that IGW amplitudes are enhanced prior to SSWs (Ratnam et al., 2004a; Wang and Alexander, 2010; Yamashita et al., 2010), and GWs are acknowledged to play a significant role in a wide range of SSW related processes (e.g. Dunkerton and Butchart, 1984; Richter et al., 2010;





Limpasuvan et al., 2012; France et al., 2013; Chandran et al., 2013; Siskind et al., 2010; Albers and Birner, 2014).

On the other hand, the majority of studies is concerned by the modulation of GWs by PWs (e.g. Cullens et al., 2015) and about the GW impact on the upper stratosphere/mesosphere region. Šácha et al. (2015) indicated a possible GW breaking in the lower stratosphere. Indeed, model experiments with gravity wave drag (GWD) parameterization showed the ability of orographic gravity wave induced drag in the lower stratosphere to significantly affect the development of sudden warmings (Pawson, 1997; Lawrence 1997) and the large scale flow in the lower stratosphere and troposphere in general (McFarlane, 1987; Alexander et al., 2010; Sandu et al., 2016).

Recently, artificially enhancing the GWD by modifications of the orographic wave sources in the parameterization McLandress et al. (2012) found that the extra GWD results in changes of PW drag. This was called a compensation process and was further statistically confirmed by Cohen et al. (2013), who interpreted it as a response of the resolved waves to maintain a ''sensible'' stable circulation. Such a response is expected, since all processes in the atmosphere are driven by the tendency to reach a more energetically favorable, stable state.

In this paper, we focus on the physical mechanism and structure of the atmospheric response to the forcing represented by an artificially injected GWD. We are following Šácha et al. (2015), who described a localized area of enhanced GW activity and breaking in the lower stratosphere over the Eastern Asia/North-western Pacific (EA/NP) region and discussed possible implications of this GW hotspot for large scale dynamics and transport. By artificially enhancing the GWD in a 3D mechanistic circulation model of the middle atmosphere (MUAM; Pogoreltsev et al., 2007) we examine the hypothesis that such a robust breaking region plays a role in forcing longitudinal variability of the BDC and can generate PWs. Further, we investigate possible implications for the polar vortex stability and the role of distribution of the GWD and of the artificial force components (direction of the force).

The structure of the paper is as follows: in section 2 we describe the model and sensitivity simulation set up together with the observational motivation and justification for an artificial GWD enhancement. The section closes with a brief description of tracer data used in this study. Section 3 starts with an illustration of a geopotential response to different GWD injections with particular focus on effects in the polar region. We also present the dynamical impact, structure and modes of the PWs generated by the artificial GWD. Finally we show the differences in BDC due to GWD geometry and analyze the 3D residual circulation spatial patterns in relation to the GWD distribution. In Section 4 we give a summary of our results, discuss potential implications of our findings and outline future directions of our work.

## 2 Data and methodology

### 2.1 Model description and configuration

We use the Middle and Upper Atmosphere Model (MUAM), which is a nonlinear 3D mechanistic global circulation model. It has a horizontal resolution of $5 \times 5.625°$ and extends in 56 vertical layers





up to an altitude of about 160 km in log-pressure height as described by Pogoreltsev et al. (2007). At 1000 hPa, the lower boundary of the model, we introduce stationary PW of wave numbers 1, 2 and 3 obtained from ERA Interim (ERAI) temperature and geopotential reanalysis data (ECMWF, 2016). Up to an altitude of 30 km the model zonal mean temperature is nudged to ERAI zonal mean temperature

to correct the climatology in the troposphere. This assimilation of stationary PWs and zonal mean temperatures is not only active during the spin-up but also during the analysis period. The time step of the model is 225 s following a Matsuno (1966) integration scheme. For simulations, the model starts with a globally uniform temperature profile and no wind. During a spin-up period, the mean circulation is built, and stationary PW and tides are added. After that, a time interval of 30 modeled days with a

temporal resolution of 2h is analyzed. We use January conditions here.

GWs are parameterized after a linear Lindzen (1981) type scheme updated as described in Fröhlich et al. (2003) and Jacobi et al. (2006) and are initialized at an altitude of 10 km with six different phase speeds ranging from 5 to 30 m/s, each propagating in eight different azimuth angles, and with GW vertical velocity amplitudes with an average value of 0.01 $ms^{-1}$. As an input for a GW parameterization

scheme, we modify a GW source function to reflect a distribution based on the mean January field of potential energy of disturbances computed from FORMOSAT3/COSMIC radio occultation density profiles between the tropopause and 35 km altitude taken from Šácha et al. (2015). The GW weights are calculated from these data by dividing the potential energy at each grid point by its global mean. This setup has a positive impact on some climatological features in MUAM. Nevertheless, the effect on

the horizontal distribution of GWD in the stratosphere is negligible. We will refer to this setup as the reference simulation. Zonal (gcu) and meridional (gcv) flow acceleration as well as the heating due to breaking or dissipation of GWs (gt) is calculated by the parameterization scheme.

To examine and to demonstrate the effect of spatial distribution of the GW activity we performed a set of sensitivity simulations (Table 1) with artificially changed GWD imposed on the model by

modulating the GW parameterization output. Note that this change of GWD is only added after the spin-up so that only the 30 model days incorporates GWD changes. Thus, the simulation period also includes the temporally delayed response for the adaption from reference conditions to enhanced GWD (gcu/gcv/gt) values.

The enhancement is performed for a certain 3-dimensional box in the lower stratosphere (about 18-30

km) above the EA/NP region (37.5-62.5°N/112.5-168.8°E), according to the area of enhanced IGW activity described by Šácha et al. (2015). This refers to the "box" distribution in Table 1 (an example is shown in Fig. 1, left panel). In a second version we additionally average the respective GWD parameters zonally within the same latitude range like the box. This way, we obtain a zonally uniform distribution, i.e. a ring of enhanced GWD parameters instead of a box but with a smaller local

magnitude. We refer to this configuration as ring or "Zon" simulations (see Table 1). For all simulations, the GWD parameters outside the box or the ring, respectively, remain unchanged and are not influenced by the enhancement. We are not smoothing the boundaries of the artificial enhancement area and the step between artificial and background GWD values is dependent on the horizontal location, the time step and, most importantly, the altitude level.





Our motivation for imposing an artificial drag stems from the findings of Šácha et al. (2015), who showed a localized area of enhanced GW activity and breaking in the lower stratosphere of the EA/NP region. In our sensitivity simulations we imitate the impact of the EA/NP lower stratospheric IGW breaking, which is not covered by the MUAM GW parameterization in the reference simulation

Although it is impossible to directly compute the GW drag force from current satellite measurements alone (Alexander and Sato, 2015), Ern et al. (2011) gave a methodology to estimate absolute values of a "potential acceleration" caused by GWs (maximum zonal mean values of 3 m/s/day below 40km). Using ray tracing simulations Kalisch et al. (2014) estimated a zonal averaged GWD to be around 20 m/s/day in the lower stratosphere. In our model simulations we are injecting three values of additional
artificial zonal component of GWD, -0.5 m/s/day as a conservative enhancement and -10 m/s/day to demonstrate a big impact of the injection. In addition, an extreme case with -70 m/s/day is added to force substantial circulation changes.

Depending on the GW type and on the direction of background winds the GWD has also a meridional component, which is usually poorly constrained by observations. We performed simulations with three
different values of meridional GW induced acceleration (-0.5 m/s/day, -0.1 m/s/day, 0.1 m/s/day).

Taking into account the prevailing directions of horizontal winds in the EA/NP region (see Šácha et al., 2015) we argue that the 3:1 ratio between the zonal and meridional GW induced acceleration is realistic and therefore we choose the Box0.1 simulation as a representative and realistic conservative GWD enhancement. A comprehensive discussion of our sensitivity simulation set-ups is given in the
Discussion section.

**2.2 Tracers**

It is difficult to evaluate model results for large-scale stratospheric vertical velocities, as these are much smaller than horizontal velocities. Therefore, the vertical structure and longitudinal variability of the residual circulation induced by the different configuration and distribution of GWD is compared with
zonal cross sections of Michelson Interferometer for Passive Atmospheric Sounding (MIPAS) $CH_4$ volume mixing ratio profiles (KIT, 2016; see von Clarmann et al., 2009; Plieninger et al., 2015). Further, to show the robustness of our claim of an enhanced branch of the BDC in the EA/NP region we computed the 1978 to 2008 average total ozone annual cycle from the ozone multi sensor reanalysis version 1 (MSR1; van der A et al., 2010) data (TEMIS, 2016).

However, the interpretation of the differences of the distributions must be done with care, since the tracer distributions result from several different processes in the atmosphere, namely advective transport, mixing, and chemical reactions (Garny et al., 2014). Also, the residual velocities are closely related to Lagrangian-mean velocities up to $O(\alpha^2)$ only for small amplitude ($\alpha$) steady waves (Bühler, 2014). We study consequences of the IGW hotspot for the longitudinal variability of the BDC by
means of the time mean 3D residual circulation according to Kinoshita and Sato (2013). The time averaging inserts additional uncertainty in the 3D residual circulation concept. In contrast to Demirhan Bari et al. (2013), who based their analysis on monthly means and daily eddies we are employing a 5-day running average on the 6 hourly MUAM output fields. For this configuration the zonally averaged




Stokes drift gives subjectively most reasonable values when compared to the transformed Eulerian mean residual circulation which is computed in this study according to Hardiman et al. (2010) for log-pressure height vertical coordinate models.

### 3. Results

#### 3.1 Atmospheric response to variations in a GWD character and SSW

Fig.2(1A) shows the monthly mean horizontal wind and geopotential field at the 6.25 hPa level (13th model level) for the Ref simulation and the remaining plots in the first row show anomalies caused by different components of GWD with artificial values corresponding to the Box0.1 simulation. The second row (Fig. 2(2A) – Fig 2.(2D)) shows horizontal wind and geopotential anomalies for the 10box (Fig.2(2A)) and Box0.1 (Fig. 2(2B)) simulations and differences between simulations with conservative GWD enhancements (Figs. 2(2C) and 2(2D)). The third row (Figs. 2(3A) through 2(3D)) shows the same as the 2nd row, but for the artificial ring GWD configuration. Note that each color bar is scaled to different values reflecting the magnitude of the differences to the reference. The SSW simulations are not compared with the others here, because they cause much stronger dynamical changes and are therefore analyzed separately.

The anomalies and differences are analyzed with special focus on the polar vortex response, since it will be shown below that the dynamical response on GWD changes is strongest in the polar region. This comparison demonstrates not only the importance of the role of the longitudinal distribution of the zonal mean drag force but it also presents a partitioning of the dynamical effect of individual GWD components.

From a comparison of Figs. 2(1B), 2(1C), and 2(1D) we see that among the artificial GWD components used in the Box0.1 simulation the zonal drag induced anomaly is strongest inducing a dipole structured anomaly with negative geopotential anomaly downwind from the region of GWD enhancement and positive anomaly northward of this region. This positive anomaly positively interferes with a smaller positive anomaly induced by the GW induced heating (Fig. 2(1D)). The negative anomaly positively interferes with a smaller anomaly induced by a meridional drag only (Fig. 2(1C)). The respective geopotential response in the Box0.1gcv and the Box0.1gt simulations seems to have quite opposite features as positive temperature forcing by GW induced mixing enhances the geopotential in the upwind and northward direction from the region and slightly lowers it above the box while artificial northward deceleration has an opposite effect. Although we used a nonlinear model, the additivity of effects of different GWD components seems to hold reasonably as can be seen from the Box0.1 anomalies Fig. 2(2B) and differences with simulations with different meridional drag only (Figs. 2(2C) and 2(2D)).

There are two important results visible from the comparison of the plots in the second and third row of Figure 2. Firstly, one may see that there are much bigger anomalies for box enhancements than for the corresponding ring enhancements. Secondly, there are significant differences (50% or 25% of the magnitude of the anomaly) between box simulations with slightly different setups of the meridional drag (Fig. 2 (2B) vs. Figs. 2(2C) and 2(2D), respectively), while this is not true for ring GWD





enhancements (few percent; see Fig. 2(3B) vs. Figs. 2(3C) and 2(3D)). However, note the fact that unlike the box enhancements, ring enhancements are almost insensitive to the different versions of GWD in the meridional direction. This could be a result of nudging, because in MUAM simulations only the zonal mean fields are nudged to the zonal mean ERAI. Therefore nudging has no direct effect

on the wave structure of the response but is likely to suppress a weak zonally symmetric response.

As noted above, the magnitude of the geopotential response is larger for the box enhancements than for the ring enhancements. For the Box0.1 simulation, which comprises a conservative GWD enhancement, the geopotential anomaly reaches about 30 gpm in the later time steps and about 20 gpm on a monthly mean at 6.25 hPa (about 35.5 km), which is chosen for presenting most of our results.

The horizontal wind anomaly for the Box0.1 simulation (Fig. 2(2B)) reaches maximal values slightly below 1m/s, which is comparable in magnitude with the projected changes under the RCP4.5 and RCP8.5 scenarios as found by He et al. (2015) at 850 hPa in the western North Pacific subtropical high region. The direct effect of the GW breaking will most likely be negligible at the 850 hPa level and we cannot investigate this, since our model has an idealistic troposphere. However, the location of the GW

hotspot in this region can be relevant to the conclusions of He et al. (2015) who found a large intermodel spread of the projected change in this region and suggested to pay attention on this region in climate change research.

Anomalies for the 10box simulation (Fig. 2(2A), 20 times bigger eastward deceleration) are almost exactly 20 times stronger and show a very similar dipole pattern. Contrary to our expectations these

simulations lead to anomalies that would contribute to weakening rather than amplification of the Aleutian high. Based on the results and discussion of Šácha et al. (2015), who argued that the EA/NP hotspot (high GW activity already in October/November) may play a role in the onset of the winter circulation in the stratosphere in this region, we expected a positive interference of the GWD response with the background climatology (e.g. contribution to the unusually hot temperatures in the

stratosphere in the EA/NP region by an induced subsidence).

The geographical location and evolution of the stationary positive geopotential anomaly together with anomalous anticyclonic horizontal winds upstream of the GWD area is a remarkable feature and can be relevant to tropospheric blocking episodes. There is statistical evidence of the relationship between SSWs and blocking patterns in the troposphere (Martius et al., 2009, Albers and Birner, 2014).

Persistent GW breaking in response to some advantageous tropospheric weather event (surface winds across topography) may be the underlying physical mechanism behind this relationship. We aim to investigate this in our future research.

Although the results presented in Fig. 2 are the mean anomalies only, while the true dynamics can be seen more clearly in a time evolution, the animations of the geopotential anomaly evolution (not shown

here) reveal that the anomalies are almost stationary, and only their magnitude is building up after the start while being steady after approximately five days after the GWD injection. No significant dynamical changes (e.g., vortex displacement) were induced in the simulations presented in the Fig. 2. Therefore we present also results from the SSWbox and SSWzon simulation, where, in response to a





strong GWD increase, we observed a vortex displacement (Fig. 3a and Animation 1a in the Supplement) and a vortex split like event (Fig. 3b and Animation 1b in the Supplement).

In the SSWbox simulation, right after the start (injection of GWD), a formation of a positive geopotential anomaly begins to form above the Northern Pacific (northward from the GWD area). This anomaly strengthens and shifts a little westward above Siberia. We observe an evolution (not only some anomaly in the difference plot) of a pressure high above Siberia and the vortex is consequently shifted towards Northern America, because the anomaly is approximately ten times stronger than in the 10box simulation. In the SSWzon simulation we also observe a slow build up of a pressure high above the Northern Pacific together with a high-pressure ridge above the Northern Atlantic. However, this pressure high is almost stationary leading to the vortex split approximately ten days after the injection. This is a potentially very interesting result suggesting that a symmetric forcing favors vortex split and localized forcing favors displacement events.

However, both events have limited vertical extent, and significantly disturb the vortex only up to 60 km (not shown). In Figures 3a and 3b the geopotential field and horizontal wind speed 300 hours after the GWD injection is shown, when the vortex split develops. The vortex displacement event is more rapid.

Subsequently, we choose the Box0.1 simulation as being representative of a conservative GWD enhancement for further analysis. Because, given the background wind fields and the presumed importance of orographic GWs in the EA/NP region in January (Šácha et al., 2015), this is the most likely direction and ratio of a decelerating force.

**3.2 Creation of planetary waves and dynamical impact**

In this section we compare PW activity and amplitude structure of the leading PW modes between reference, Box0.1, and Zon0.1 simulations. We show results of the E-P flux diagnostics and Fourier transform (FT) analysis of geopotential height and zonal wind zonal anomalies.

Fig. 4 shows the monthly mean E-P flux and its divergence for the Box0.1 simulation (Fig. 4a), anomalies with respect to the reference simulation (Fig. 4b), Zon0.1 simulation anomalies with respect to the reference simulation (Fig. 4c), and the difference between the Zon0.1 and Box0.1 simulations (Fig. 4d). Note that we show the E-P flux divergence as a force per unit area (units $[kgm^{-1}s^{-2}]$), not as an induced acceleration (units $[ms^{-2}]$), as in Hardiman et al. (2010), because the differences or anomalies would be dominated by upper stratospheric and mesospheric effects due to density decrease.

In Fig. 4b, for the Box0.1 simulation, we find an anomalously weak E-P flux convergence (positive difference to the Ref simulation) centered at the equatorward flank of the GWD enhancement area and an anomalous convergence in a broad area around 60°N. This pattern is similar in the Zon0.1 simulation, but much weaker, and in both simulations it is limited in altitude by the upper boundary of the GWD area. In line with the above discussed fact that we are plotting force per unit area, we can observe the strongest E-P flux differences in the lower part of the GWD enhancement area.

Putting together the anomalies in Fig. 4b with a full field from the Ref simulation (Fig. 4a), we see that the Box0.1 anomalies are due to a stronger poleward and also upward propagation of PWs along the




northern edge of the jet from the northern part of the GWD area, and due to a suppression of upward and equatorward propagation elsewhere in the GWD region. South of 30°N only, there is a small area of anomalous convergence due to the anomalous equatorward propagation from the GWD area. For this conservative Box0.1 simulation the differences in the E-P divergence are around 5% of the

reference values. In contrast to that, Zon0.1 E-P divergence anomalies reach only 1 or 2% of the reference values, locally. These results confirm that the localized forcing has a potential to modify PW activity more efficiently than a zonally uniform drag of the same zonal mean zonal force.

Focusing on the mean fields from the 10box simulation (Fig. 4e) we may observe pronounced anomalous equatorward propagation as well. Fig 4e also demonstrates the mechanism behind the

anomalies (Fig. 4b) of the conservative Box0.1 simulation. Anomalies of the 10box simulation (not shown) exhibit the same pattern, but the magnitude is much stronger - more than 50% of the original E-P flux and divergence values. Therefore we observe an influence of the GWD box enhancement also in the mean fields. The artificial GWD box demonstrates itself as an E-P flux divergence area on the southern flank of the GWD enhancement region focusing the PWs in its northern part to propagate

vertically and northward towards the jet core and along the northern edge of the jet and restraining PW propagation upward and equatorward along the southern vortex edge.

On the other hand, the box GWD enhancement creates more horizontally equatorward-propagating PWs contributing to the E-P convergence in the tropics. Presumably, the box enhancement creates also poleward propagating PWs, but they are not visible in the much stronger reference E-P flux fields

northward from the region. This is more apparent, together with the alteration of the jet in the lower stratosphere (indication of this is visible also when comparing Figs. 4e and 4f), from Fig. 1S in the Supplement. Here we present E-P flux and its divergence at particular time moments, especially in the first model hours after the GWD injection for the SSWbox simulation. In Fig. 1S we can see that the structure of the E-P flux divergence area (in the bottom part and below the GWD area) changes with

time and also the propagation directions of PWs created in this region are time dependent. In this simulation with the strongest GWD injection made, the E-P flux convergence decelerates the westerly zonal mean zonal wind in the polar region below 35km. This is probably a result of zonal averaging of the displaced vortex and the anticyclonic wind fields (see Fig. 3 and related discussion) and it is a clear signal that a strong artificial drag can overcome the nudging effect.

FT analysis of zonal geopotential and zonal wind anomalies provides information about the representation of different harmonics in the anomalous wave activity revealed in Fig. 4, and about the spatiotemporal distribution of their amplitudes. From the time evolution of total harmonic amplitude anomalies (difference with the amplitudes from Ref simulation) for Box0.1 simulation and in particular for the wave number s=1 anomalous amplitude (Fig. 5b), we see that the time evolution of anomalous

PW activity is dependent on the relative position to the GWD region in the meridional direction (compare the red and blue curves in Fig.5) and that there is a peak of a sum of anomalous wave activity amplitude around 6 days after imposing the artificial GWD. For the s=1 harmonic amplitude the peak





appears later (around day #14) at 62.5°N while equatorward from the enhanced GWD region the time evolution is almost constant (Fig. 5b, blue curve for 27.5°N).

An oscillating pattern in the time evolution of the sum of wave activity and in the time evolution of s=1 harmonic amplitude presumably originates from inertia GWs (period of one day for s=1) that are constantly radiating from the region of artificial GWD. On the other hand we assume that the bulk FFT features (without the oscillatory pattern) stand for the PW activity.

We do not consider the occurrence of inertia GWs, which originate from the artificiality of the GWD injection in our simulations, to be an undesirable result only, because in reality this can also be the case as the influence of GWs on the background comes from a sudden and unbalanced breaking of wave packets. Those inertia GWs are probably responsible for propagation of the anomalous wave activity through the Rossby wave critical layer in the tropics, across the equator, and into the Southern Hemisphere (SH, see Animation 2 in the Supplement).

In Fig. 6 the mean latitudinal structure of amplitudes of different PW modes in the Box0.1 simulation (Fig. 6a,b) is shown together with their anomalies from the reference simulation (Fig. 6c,d) and differences from the Zon0.1 simulation (Fig 6e,f). Because the time evolution of the anomalous activity is oscillating at shorter time scales and is almost periodical on longer scales, we take this monthly mean latitudinal structure as representative for the period of simulation.

The wave-1 geopotential amplitude anomaly between box and reference and the difference between box and ring simulation reach a maximum around the northern side of the polar vortex edge, and a minimum around the northern part of the GWD area. We can find maximal anomalous amplitude at a similar latitude also for wave-3, although slightly northward from the GWD region it reaches a secondary maximum. The box configuration obviously enhances odd planetary wave modes in comparison to the reference and also the ring GWD configuration. In the Box0.1 simulation, wave-2 has a pronounced negative anomaly at the northern edge of the enhancement region, with similar value of difference with the Zon0.1 simulation.

We are enclosing an animation of the time evolution of the latitudinal structure in the Supplement (Animation 2), where one may see the slow buildup of anomalous wave amplitudes due to the PW activity northward from the GWD area and the propagation of anomalies into the SH, and pulsation of the amplitude maxima due to inertia GWs.

Time evolution and the latitudinal structure of the PW amplitudes were presented on the 13th model level corresponding roughly to 35km log pressure height (upper half of the GWD enhancement area), which is slightly above the level of possible GW breaking as observed by Šácha et al. (2015). Although we have found the largest differences in the PW activity around 20 km using the E-P diagnostics, for a FT analysis we decided to choose a level slightly above the GWD area. In a supplementary analysis of the vertical structure of anomalous harmonic amplitudes (see Animation 3 in the Supplement), it is shown that the amplitudes of geopotential anomalies are largest in the upper stratosphere (taking the mesosphere and thermosphere not into account) and for the wave-1 the maxima are descending in time to altitudes below 30 km.





In the subsequent evolution, around 16 days after the GWD injection and at an altitude of 45 km, a PW reflection-like behavior of the wave-1 anomalies occurs. At this time, we observe a formation of a secondary maximum around 35km and a weakening of the wave-1 amplitude corresponding to Fig. 5b. This reflection is not as pronounced for the wave-2 anomalies and is not present for the wave-3

anomalies that propagate freely to the upper levels, forming a single anomalous amplitude maximum, but at lower levels (around 40 km) than wave-1. It remains unclear what this means in relation to the E-P flux results where we have found the biggest anomalies in the lower stratosphere. But these results are not directly comparable, as the E-P flux is not scaled with respect to the density decrease (to demonstrate the in-situ forcing in the lower stratosphere), while the wave amplitude anomaly is

expected to grow with height.

**3.3 Residual circulation response**

Fig. 7 shows the residual circulation mass fluxes, anomalies and differences for the Box0.1simulation on the left and 10box simulation on the right. There are some very remarkable results to see. First, even for a conservative drag enhancement (Box0.1 simulation) there are differences in the magnitude of the

residual mass flux between box and ring GWD distribution up to 3% in the lower stratosphere (Fig. 7e). For the 10box simulation the differences reach about 40% and create a similar pattern as for the conservative enhancement (Fig. 7f). The largest differences between the two configurations are found poleward from the GWD enhancement region in the altitude range between 20 and 30 km corresponding approximately to the vertical extent of the area.

The differences are slightly weaker southward from the enhancement region constituting together a butterfly like pattern in the box-ring differences centered at approximately 45°N (the center of the enhancement region) and influencing a shallow BDC branch. Taking into account the full fields (see Fig. 7b) we can explain this pattern by a faster northward advection starting at approx. 45°N and stronger subsidence northward of 60°N. On the other hand there is less upwelling in the equatorial

region and slower advection from the tropics. The continuity is satisfied through smaller downwelling south of 60°N. We observe a similar but stronger pattern in the anomalous fields with the zonal mean residual circulation mass flux anomaly reaching up to 5% for the Box0.1 simulation and more than 60% for the 10Box simulation. This is in agreement with the E-P flux divergence anomalies shown in Fig. 5. In the lower layers of the artificial GWD area the Box0.1 simulation has a weaker E-P flux

divergence at the southern boundary and a stronger one north of the center.

In the anomalous fields (Figs. 7c and 7d) the butterfly like pattern is centered more southward (35°N) than in box-ring differences and the southward anomalous pattern is weaker in magnitude and not as well pronounced. In the upper stratosphere there are differences and anomalies up to 2% only for the Box0.1 and locally around 25% for the 10box simulation. The box simulations show weaker

subsidence towards the polar vortex center than the Ref simulation in the upper stratosphere and there is also anomalously low flux poleward and downward between 30 and 40 km of height approximately above the GWD enhancement region. The differences between box and ring simulations (Fig. 7e, f) give a weak hint of the upper BDC branch acceleration for localized GWD compared to a zonally uniform distribution of artificial GWD.





In the SH, the differences between Box0.1 and Zon0.1 occasionally reach a few percent. This, together with the previously unexpected fact of a weak upper branch BDC response to the artificial GWD can be rather explained by the effect of a monthly averaging than due to the locality of the residual circulation response to the artificial GWD (see Animation 4 in the Supplement). At particular time steps the magnitude of the anomalies is comparable regardless on the BDC branch, but for the deep BDC branch the anomalies oscillate.

Animation 4 presents the time evolution of the zonal mean residual circulation associated mass flux for the 10box simulation together with its anomaly to show the global nature of the response and to give insight into how quickly the residual circulation gets affected by the NH anomalous forcing. Soon after the artificial GWD injection two anomalous residual circulation branches evolve in accordance with theory (Haynes et al., 1991) - one of them stronger below the upper boundary of the GWD layer and the one weaker above the upper boundary. The upper anomalous branch is constituted by anomalously poleward moving air at about 30 km height, which is consequently not subsiding in the polar vortex but moves upward and equatorward. Such an anomalous residual circulation pattern leads to a quadrupole like distribution of anomalies of quantities of state as discussed, e.g., by Kuchar et al. (2015).

In Animation 4 (right panel), right after the start one may also see the first signs of anomalies in the SH, namely at the South Pole. As this would require propagation of information faster than the speed of sound in the atmosphere, it is almost certainly not of a physical origin. The first response at the South Pole that can be considered to be physical emerges around 16 hours after the injection and corresponds to a propagation of information near the speed of sound. Also in the further evolution, the anomalies propagate southward in a wavelike manner, whereas there is one constant anomaly corresponding roughly to an accelerated shallow BDC branch sloping down from approx. 30km at the North Pole to the lowest levels at the equator. Except for this region, the entire domain is dominated by anomalies seemingly descending downward from the mesosphere causing the residual wind anomaly to oscillate with a 24h period (Animation 4, right panel). Thus, we once have anomalous upwelling and once downwelling at the southern polar region with the phase transition by $\pi$ around 42km log-pressure height.

From the time evolution of the zonal mean residual circulation mass flux in the 10box simulation (Animation 4, left panel) one may see that the artificial GWD region acts like an obstacle for northward flowing wind creating a lee wave like pattern. This may explain the ostensible downward propagation of anomalies discussed above. Further developing this analogy, the GWD enhancement (an obstacle) is steadily flown around inducing anomalous upwelling on its southern edge and downwelling on the northern flank.

The zonal structure of the induced flow, and possible consequences of the GW hotspot for the longitudinal variability of the BDC were studied by means of 3D residual circulation analysis according to Kinoshita and Sato (2013). 5-day running averaging is performed. The averaging period was subjectively chosen on the basis of a series of tests as giving the most reasonable zonally averaged Stokes drift when applied to the Box01 simulation.

Šácha et al. (2015) pointed out unusually high temperatures in the EA/NP region at 30 hPa in winter





and concluded that there could be an enhanced downwelling above the EA/NP region penetrating to lower levels than elsewhere. This is in agreement with Fig. 3 in Demirhan Bari et al. (2013). To demonstrate the robustness of the enhanced BDC branch in this region, in Fig. 8 we present a thirty-year average January MSR total ozone column field.

In Fig. 8 we can see a significant total ozone column maximum corresponding to the EA/NP region. When presenting total column ozone one has to have in mind the westward shift of the subsidence area with height (see Fig. 9, upper row) and take into account the climatological height profile of ozone concentrations. Mean January zonal cross-sections of MUAM vertical residual velocity for the reference simulation (Fig. 9, upper row) give insight into the zonal structure of the residual circulation.

A good correspondence between the location of the maximal subsidence branch in Fig. 9 and the location of the total ozone column maximum (having the above discussed prerequisites in mind), shows that the enhanced downwelling pattern in the EA/NP region is well captured in MUAM.

Northward of approximately 40°N (not shown), MUAM residual vertical velocity field is dominated by a wave-2 pattern, with the maximum subsidence branch penetrating to the lower stratosphere in the

EA/NP region. Ridges and troughs of the wave show a characteristic westward tilt with height. The comparison of the residual vertical velocity distribution with MIPAS $CH_4$ volume mixing ratios (lower row of Fig. 9), shows especially good agreement in the location of subsidence peaking around 15 km at 140°E and the consequent massive upwelling branch east of it. The other two wave-2 peaks are weaker in the residual vertical velocity field and not well pronounced in the $CH_4$ cross-sections. Indications of

this pattern can be seen in the $CH_4$ distribution only around 20 km altitude, otherwise the tracer distribution is dominated by a wave-1 pattern.

To analyze the role of GW activity in this longitudinal variability of the BDC we show longitudinal cross-sections of the 3D residual vertical velocity anomalies in Fig. 10. The anomalies are computed for the Box0.1 simulation which was discussed to have the most realistic direction of the GWD for the

EA/NP region. As this drag enhancement is conservative and the induced dynamical changes are small, we consider the presented GW induced anomalous residual vertical velocity patterns to be representative and universal for this kind of drag.

Šácha et al. (2015) hypothesized that the collocation of the EA/NP GW hotspot and the enhanced BDC branch can be partly a consequence of the circulation induced by GW breaking. But from Fig. 10 we

can see that in reality this is not as straightforward. At the equatorward flank of the area, the GWD induces anomalous upward flow exclusively, confirming results of the zonal mean residual circulation analysis. Anomalous subsidence begins to be significant further northward where, in line with the obstacle analogy, we observe subsidence in the eastern part of the GWD region only, while anomalous upward flow dominates the western part of the GWD region, and then again eastward and slightly

above the anomalous subsidence area.

This unexpected result shows that not only the downwelling, but also upwelling patterns may be related with GWs. The magnitude of the anomalies is around 10% of the climatological value. Physically, such





an anomalous pattern can make sense if we consider the dominant background horizontal north-eastward wind together with the previously mentioned small obstacle analogy with upward flow upwind and downward flow downwind from the GWD box. On the other hand, for the SSWbox simulation we can observe counterintuitive subsidence located directly above the GWD area, except for

5 the northern edge of the enhanced GWD region (see Supplement Fig. 2S), but the global pattern is very different from expectations (wave-1 or wave-2 pattern in reality vs. strong uplift at the eastward side of the area in the Fig. 2S). When the artificial GWD is strong enough to induce significant dynamical changes (SSWsimulation) the anomalies cannot be directly explained as being GW induced because also the dynamical state of the atmosphere changes (e.g. the anticyclonic evolution in Animation 1a).

10 Therefore, the explanation of residual vertical wind cross-section patterns for both SSW simulations is much more complicated and requires future research allowing at least the GWD enhancement to reflect the changing background conditions.

## 4. Discussion and conclusions

We will begin this section discussing the discrepancy of the subsidence pattern between different GWD

15 set-ups (Box0.1 and SSW simulations). These are probably due to the induced dynamical changes in the model state of the atmosphere in the SSW simulation. Also, a contrasting effect of the different GWD components and their different ratios may to some extent determine the induced residual circulation pattern, e.g. the heating induced by GW breaking related mixing - positive in our case - induces a positive geopotential anomaly locally, while the meridional component of GWD is connected

20 with a negative anomaly (See Fig. 3).

It is important to focus on the residual velocity anomalies in future research to analyze the possibility of a positive feedback mechanism between GWD breaking and subsidence. Due to the anomalously high temperatures in the lower stratosphere in the EA/NP region, upward propagating GWs more favourably saturate and reach instability there. As a result, the GW breaking is likely to start rather low

25 in the stratosphere in this region. Due to energy loss when breaking, GWs most certainly produce some direct heating and it is more complicated with the induced mixing that can result both in heating or cooling (Liu et al., 2000). A question to answer in future is, how the circulation induced by GW breaking looks like and to what extent the occurrence of a persistent GW breaking can contribute to local subsidence. However, such a two-way feedback cannot be assessed by artificial injection of

30 GWD.

The artificiality of our GWD enhancement is naturally the biggest limit of our analysis when comparison is made with observations. Our GWD enhancement introduces an additional artificial constant momentum sink in the model. The concept of artificial GWD enhancement leaves us also no chance to reflect any feedback between GWs and background conditions, like the previously discussed

35 induced vertical movements or the evolving PW field. Therefore, for example, our simulation of a vortex displacement is not very reliable as the GWs are known to be significantly filtered during SSWs (e.g. Holton, 1983; Limpasuvan et al., 2012). Also, the injection of the artificial GWD is responsible for creation of inertia GWs. In our results we always pointed out when a particular feature was caused by inertia GWs, because they are a result of our sensitivity simulation set-up. Nevertheless, we believe


that it is interesting to describe these features (propagation across the equator), as it is possible that inertia-GWs indeed accompany GW breaking in reality.

Considering the intermittent nature of GWs (e.g. Hertzog et al., 2012; Wright et al., 2013), another inaccuracy of our sensitivity simulation set-ups arises from the constancy of the artificial GWD. A multiple (during a month) pulse like injection of the artificial GWD would be arguably more realistic, but on the expense of massive inertia GW occurrence during the whole simulation. It is also a question, what is a more realistic illustration of the GW effect on the atmosphere, a sudden GWD injection or smooth increase and decrease with e.g. a 10-day e-folding time to minimize the initial adjustment noise as proposed by Holton (1983)? Also the spatial distribution of our artificial GWD is highly idealized (in both the horizontal and the vertical). We must note that we compare two "extreme" GWD longitudinal distributions only.

In this study, partly because our GWD is an artificial momentum sink in the model, we did not analyze our results with regard to the recently described quantitative compensation mechanism between GWD and drag induced by resolved waves (Cohen et al., 2013). Nevertheless, our results can shed light on the physical mechanism behind the compensation, and this is the enhanced PW creation by the localized GWD area. Our results suggest that the rate of compensation can be variable in dependence on the GWD distribution that influences the efficiency of PW creation.

In future work it is necessary to take into account more realistic GWD distributions to address the efficiency of PW creation. For example, it is possible that a configuration of GWD taking into account the EA/NP and e.g. the Greenland GW activity hotspot would favour enhanced wave-2 instead of wave-1 activity, and for comparison a chess like or random distribution of GWD would possibly be more appropriate for comparison. Another motivation for future research is to analyze the role of the GW hotspot position relative to the climatological stationary wave location in the stratosphere, with special focus on the position of the EA/NP hotspot in the region of the transition phase between a trough and ridge of climatological wave-1.

Watt-Meyer and Kushner (2015) argued that standing waves drive the most persistent part of the wave activity flux suggesting that these should be a primer connection between the wave driving and stratospheric polar vortex strength, and e.g. Yamashita et al. (2015) found constructive interference leading to the wave-1 amplification corresponding to the weakened polar vortex in late winter during the westerly phase of the equatorial quasi-biennial oscillation under solar maximum conditions. Our results clearly show (Fig. 6) an anomalous amplification of wave-1 amplitude for a box GWD enhancement in the EA/NP region.

Kren et al. (2016) found that during the Pacific Decadal Oscillation (PDO) SSWs are more frequent compared to the negative phase, confirming the impact of North Pacific anomalous circulation on the wintertime polar vortex (e.g. Woo et al., 2015; Kidston et al., 2015). Woo et al. (2015) attributed the differences between PDO phase effects on the polar vortex strength to the PDO-induced tropospheric circulation anomalies over the North Pacific and consequent interference (constructive or destructive) of the anomalous wave-1 component with climatological PWs. Regardless on our previously discussed



results on the wave-1 enhancement, we may argue for the mechanism of variable sourcing, propagation and breaking conditions for GWs influencing the occurrence and strength of the EA/NP GW breaking region in dependence on the PDO phase and consequent anomalous in-situ generation of PWs in the lower stratosphere.

In the atmosphere, the most natural, immediate and fastest way for communication of information in the vertical are the GWs (apart from acoustic and acoustic-gravity waves with effects much higher in the atmosphere). Hypothetically, sea surface temperature PDO induced anomalies leading to anomalous surface winds (or convection) and consequent anomalous orographic (or convective) GW creation can immediately influence the distribution of GW breaking (not only) in the lower stratosphere
with possible in-situ lower stratospheric effects as shown in our results. This does not need to work for the PDO only, a similar mechanism can also be behind e.g. the West Pacific teleconnection pattern or others.

Regarding the vertical communication of information, Müller et al. (2015) recently presented a new theory for diagnostics of blocking episodes together with the polar vortex stability. The approach is
based on the observational evidence that such events are often connected by two or three isolated vortices represented by high over low or omega-blocked weather patterns. Blocking connection with SSWs is a well-known correlation (e.g. Andrews et al., 1987; Martius et al., 2009; Nakamura et al., 2014) but the mechanisms standing behind are still rather elusive. In future, we would like to study the analogy with our results in view of the fact that the localized GWD can create anticyclonic and
cyclonic anomalies (Fig. 3) in the lower stratosphere influencing the total sum of circulation and propagation of these patterns.

Another implication of our results may be the physical justification for disturbing the vortex in its central levels that was a mechanism hypothesized by Scott and Dritschel (2005). Traditionally, PWs are thought to be generated in the troposphere and propagate up on the polar vortex edge. But, as Scott
and Dritschel (2005) pointed out, when wave amplitudes become large and nonlinear effects become important, the notion of upward propagation ceases to be appropriate. Therefore, they considered an option of some in situ disturbance at a given level, with a possible explanation from what we propose - localized GW breaking implicating anomalous PW activity.

It is common methodology (see e.g. Albers and Birner, 2014 for a review of SSW preconditioning
concepts) to estimate e.g. the relative impact of GWs and PWs on polar vortex preconditioning from zonal mean values of zonal forces only. But our results show that the dynamical effect of forcing depends also on its distribution. The impact connected with a localized area connected with a higher value of drag can be much higher than one would expect from the zonal mean value only. Importantly, we have found that for a sufficiently strong artificial zonal mean zonal force there is a vortex split
response to the ring artificial GWD configuration and vortex displacement for a localized forcing. We aim to investigate this in more detail and also for more realistic forcing distributions, but it seems to be clear that the SSW type may be determined also by the geometry of the forcing, not only by the vortex





geometry. On the other side, vortex geometry can to a large extent influence the distribution of the forcing, e.g. spontaneous emission processes connected with the jet (Plougonven and Zhang, 2014).

Another result of our study is the role of the meridional GWD component, especially for the polar vortex response. Interestingly, this feature becomes apparent for the localized enhancement only and has an almost negligible effect in simulations with ring enhancements. To our knowledge, the effect of the meridional component of GWD on the middle atmospheric circulation has not been studied yet. Also, horizontal GW propagation is neglected in most climate model parameterizations (Kalisch et al., 2014). Thus, it is not surprising that there are only few modelling constraints regarding the horizontal propagation directions, although some information is available from ray tracing simulations (Preusse et al., 2009). In most studies based on satellite data, GW propagation directions have not been analyzed, because the information needed for such computation (hodograph analysis etc.) is not available for most of the instruments and their combinations (Wang and Alexander, 2010).

A general conclusion of this paper is that for the same magnitude of an artificial zonal mean zonal force (zonal mean meridional force as well) there are significant differences (depending on the magnitude of the GWD enhancement) in the zonal mean residual circulation and E-P flux divergence between different distributions of this force (localized vs. zonally uniform). This is a clear signal that e.g. in the research of future BDC changes from climate models we need to be concerned not merely by the magnitude or latitude-height profile of the zonal mean GWD but also by its zonal distribution. In particular, the models should be able to mimic the main GW activity hotspots. This suggests the need for improvement especially of the nonorographic GW parameterization (though nonorographic GW are usually assumed to have significant effect higher than in the vertical range analyzed in this paper), since many global climate models use e.g. a globally uniform gravity wave source function (Geller et al., 2013).

**Code availability**

MUAM model code is available from the authors upon request.

**Data availability**

MIPAS $CH_4$ volume mixing ratio profiles have been provided by Karlsruhe Institute of Technology (KIT), Institute of Meteorology and Climate Research - Atmospheric Trace Gases and Remote Sensing through https://www.imk-asf.kit.edu/english/308.php. MSR total ozone is available through ESA, Tropospheric Emission Monitoring Internet Service (TEMIS) on http://www.temis.nl/protocols/o3field/o3mean_msr2.php. ERA-Interim temperatures and geopotential heights data have been provided by ECMWF through http://www.ecmwf.int/en/research/climate-reanalysis/era-interim.

**Acknowledgements**

This study was supported by GA CR under grant 16-01562J and by Deutsche Forschungsgemeinschaft under grants JA 836/30-1 and 836/32-1.



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




| Name | Distribution of the artificial GWD | Artificial gcu per gridpoint of the artificial area (m/s/d) | Zonal mean gcu in the altitude of artificial GWD (m/s/d) | Artificial gcv per gridpoint of the artificial area (m/s/d) | Zonal mean gcv (m/s/d) | Artificial gt per gridpoint of the artificial area (K/d) |
|---|---|---|---|---|---|---|
| *Ref* | ~ | ~ | *0.011* | ~ | *-0.001* | ~ |
| *Box0.5* | *box* | *-0.5* | *-0.073* | *-0.5* | *-0.085* | *0.05* |
| *Zon0.5* | *ring* | *-0.073* | *-0.073* | *-0.085* | *-0.085* | *0.05* |
| *Box0.1pos* | *box* | *-0.5* | *-0.073* | *0.1* | *0.018* | *0.05* |
| *Zon0.1pos* | *ring* | *-0.073* | *-0.073* | *0.018* | *0.018* | *0.05* |
| *Box0.1* | *box* | *-0.5* | *-0.073* | *-0.1* | *-0.016* | *0.05* |
| *Zon0.1* | *ring* | *-0.073* | *-0.073* | *-0.016* | *-0.016* | *0.05* |
| *Box0.1gcu* | *box* | *-0.5* | *-0.073* | ~ | *-0.001* | ~ |
| *Box0.1gcv* | *box* | ~ | *0.011* | *-0.1* | *-0.016* | ~ |
| *Box0.1gt* | *box* | ~ | *0.011* | ~ | *-0.001* | *0.05* |
| *10box* | *box* | *-10* | *-1.706* | *-0.1* | *-0.016* | *0.05* |
| *10zon* | *ring* | *-1.706* | *-1.706* | *-0.016* | *-0.016* | *0.05* |
| *SSWbox* | *box* | *-70* | *-12.018* | *-0.1* | *-0.016* | *0.05* |
| *SSWzon* | *ring* | *-12.018* | *-12.018* | *-0.016* | *-0.016* | *0.05* |

**Table 1: Sensitivity simulation names and GWD settings for zonal wind drag (gcu), meridional wind drag (gcv) and heating due to GW (gt) within the box. Note the gcu enhancements are negative because the drag is westward directed. The distribution describes whether the artificially enhanced GWD is implemented only for certain longitudes (Box) or zonally uniform (Zon). The tilde "~" indicates that values are unchanged w.r.t. the reference simulation.**

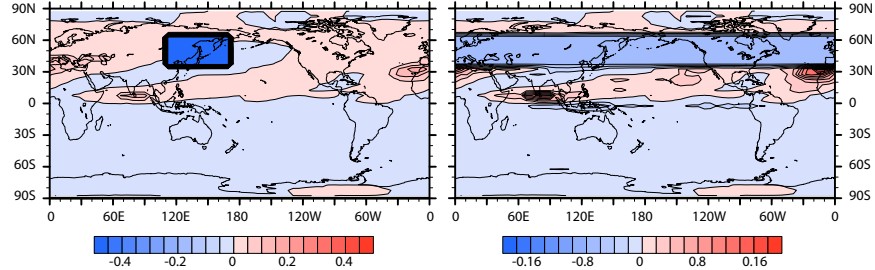

**Fig. 1 Two examples of the GWD enhancement horizontal distribution. Left panel: box distribution (Box0.1 simulation). Right panel: ring distribution (Zon0.1 simulation). Colors indicate GW induced zonal acceleration [m/s/day].**





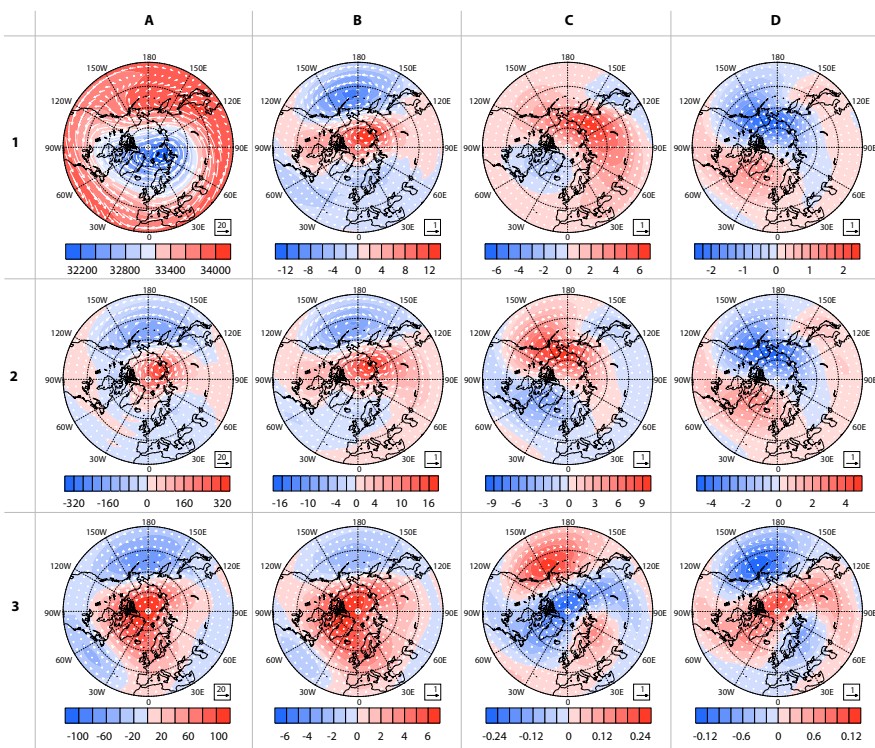

**Figure 2: Monthly mean geopotential and horizontal wind vectors at the 13th model level (6.25 hPa) for the reference simulation and differences for the sensitivity simulations with different GWD set-up. From top left (index 1A) to bottom right (index 3D): 1A) reference simulation, 1B) reference-Box0.1gcu, 1C) reference-Box0.1gt, 1D) reference-Box0.1gcv, 2A) reference-10box, 2B) reference-Box0.1, 2C) Box0.1-Box0.5, 2D) Box0.1-Box0.1pos, 3A) reference-10zon, 3B) reference-Zon0.1, 3C) Zon0.1-Zon0.5 and 3D) Zon0.1-Zon0.1pos. Colors indicate geopotential height [gpm]. Note the different scaling of the respective plots. Arrows refer to horizontal wind [m/s] with unity arrows given below the individual plots.**





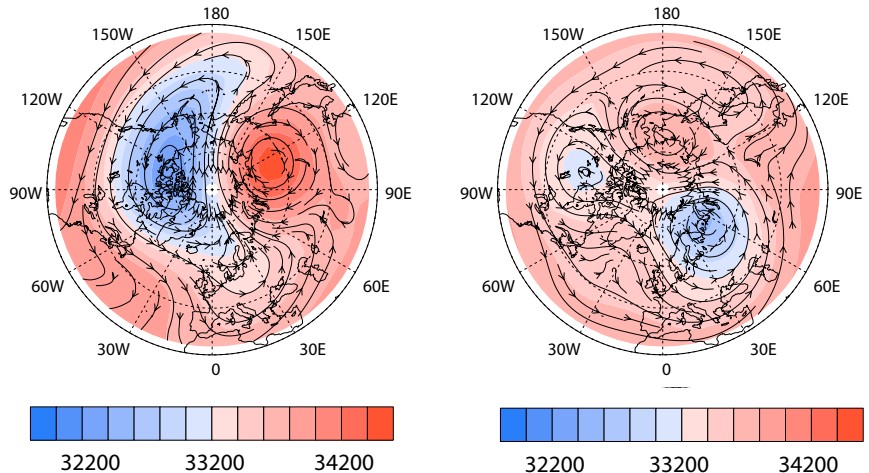

Fig. 3. **Geopotential (colors, given in gpm) and horizontal winds (stream lines, given in m/s) for the SSWbox (left) and SSWzon (right) simulation at the 13th model level (6.25 hPa) at 300 hours after the injection.**





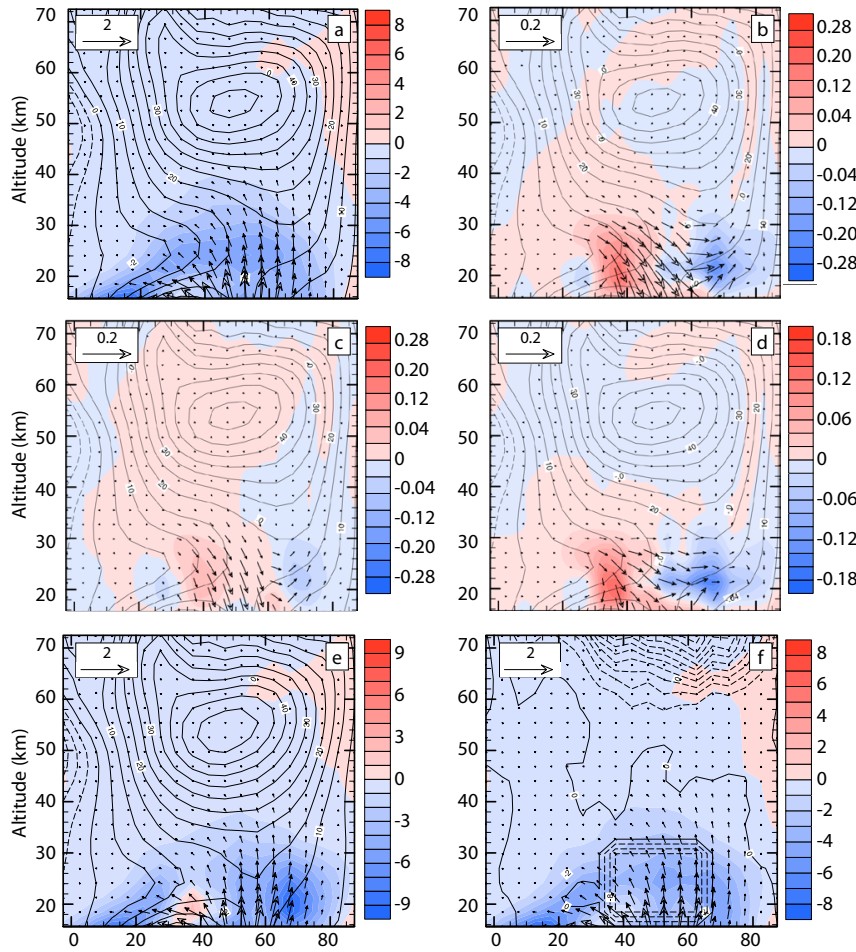

**Fig. 4.: Mean January E-P flux vectors (kgs⁻², arrows are scaled according the relative distances of the plot) and its divergence (colors in kgm⁻¹s⁻²) for Box0.1 (a), its anomalies (b), Zon0.1anomalies (c), difference between the Box0.1 and Zon0.1 simulation and mean E-P flux and its divergence for 10box (e) and 10zon simulation (f). Note that scales are adjusted for each subfigure, except the plots of anomalies (b,c) and mean fields for 10- simulations (e,f) that both share the same scaling. In panels (a-e) contours of zonal mean zonal wind from the respective simulations are overlaid with an increment of 10 m/s. In panel (f) contours of gravity wave induced zonal acceleration are overlaid to illustrate the location of artificial GWD.**





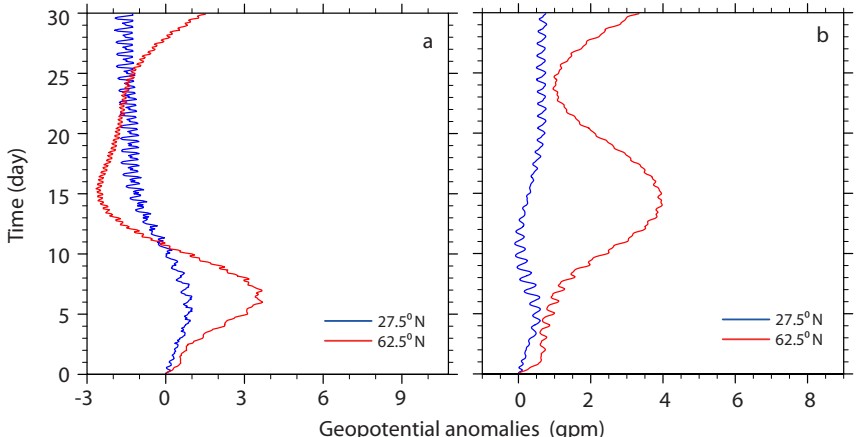

Fig.5: Time evolution of a sum of all harmonics' amplitude difference with respect to the reference run (a) and time evolution of the wave-1 amplitude difference (b) as given by the FT of geopotential at approximately 35km log pressure height for Box0.1 simulation minus FT amplitudes from Ref simulation at selected latitudes: 62.5°N (red) and 32.5°N (blue). Units are given in [gpm].

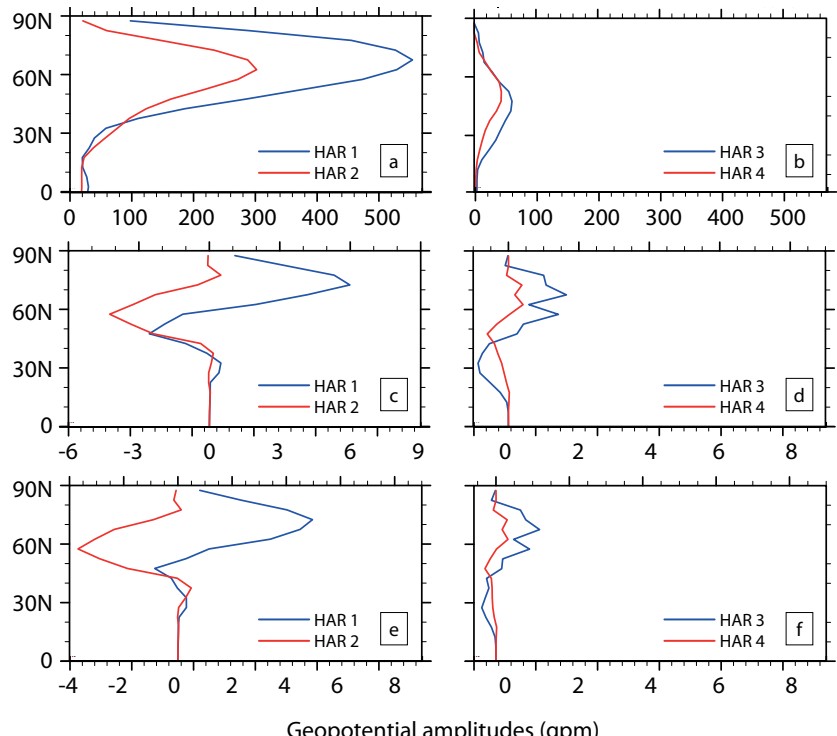

Geopotential amplitudes (gpm)

**Fig. 6: Monthly mean latitudinal structure of amplitude of selected harmonics for Box0.1 simulation. From top left to bottom right: a) harmonics 1 and 2 for Box0.1, b) harmonics 3 and 4 for Box0.1, c) differences of a) from the reference simulation, d) differences of b) from reference simulation, e) differences of a) from Zon0.1, f) differences of b) from Zon0.1. At approx. 35 km log-pressure height. Units are given in [gpm].**





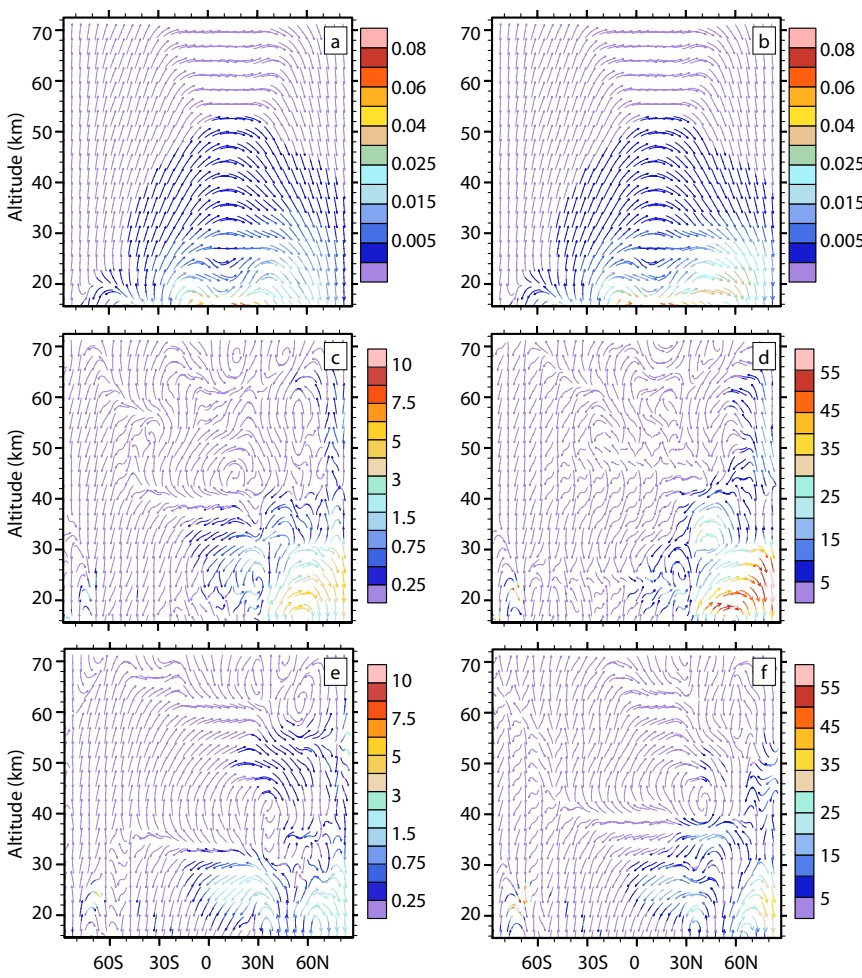

**Fig.7: Mean January zonal mean residual circulation (stream lines for illustration of direction only) and its mass flux (colors, in $kgm^2s^{-2}$) on the left (from top to bottom: Ref simulation (a), relative Box0.1- Ref simulation anomaly (c), relative Box0.1 - Zon0.1 simulation difference (e)) and on the right (from top to bottom: 10box simulation (b), 10box - ref simulation relative anomaly (d), relative 10box - 10zon simulation difference (f)). Relative anomalies and differences are given in % of the reference or corresponding box simulation, respectively.**





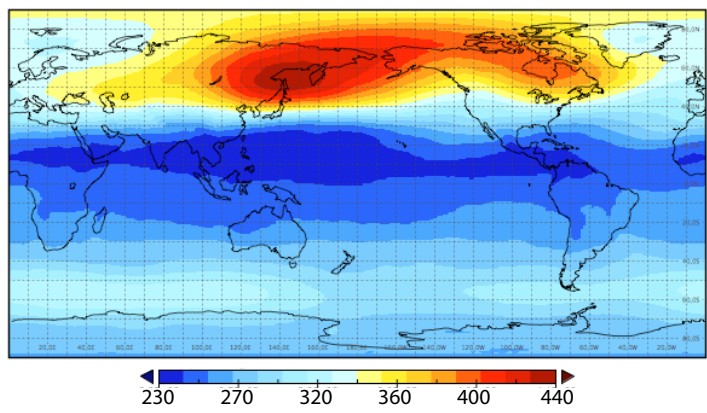

Figure 8: 1978 to 2008 mean January mean total ozone column field from MSR (units in DU).





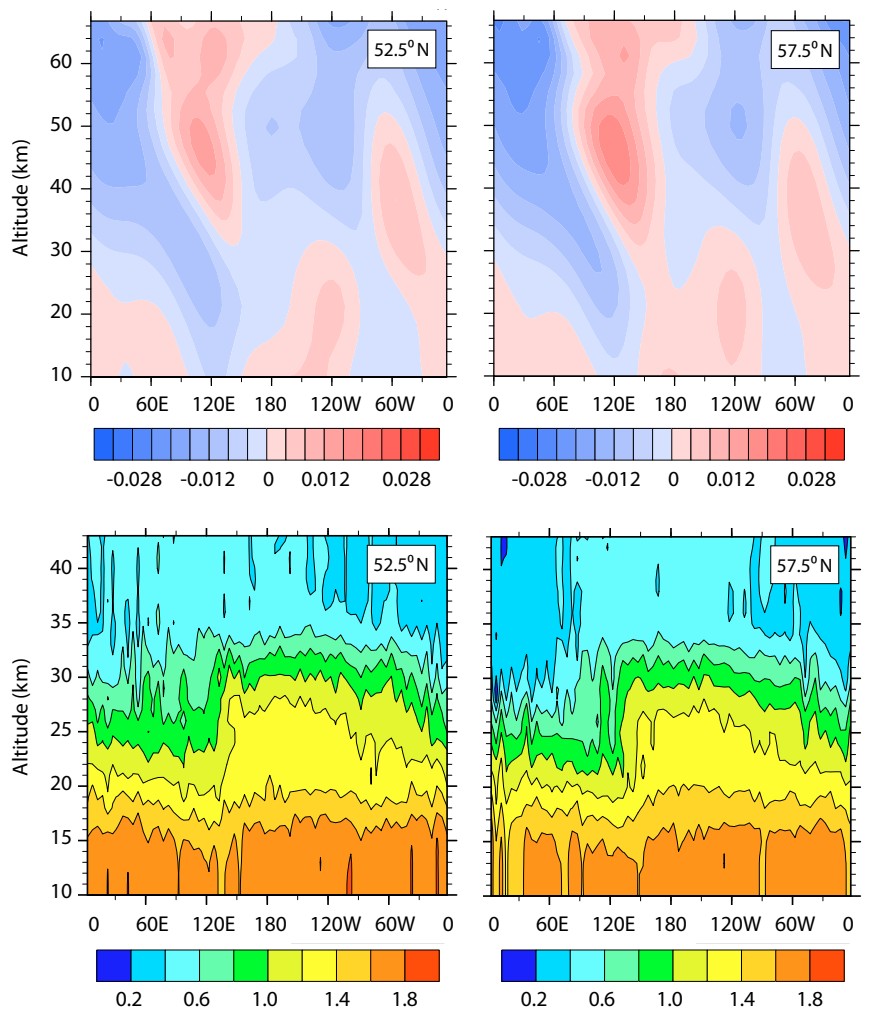

Fig. 9: Mean January zonal cross-sections of MUAM reference simulation residual vertical velocity (upper panels) and
MIPAS $CH_4$ volume mixing ratio in January 2010 (lower panels) at 52.5°N (left panels) and 57.5°N (right panels).





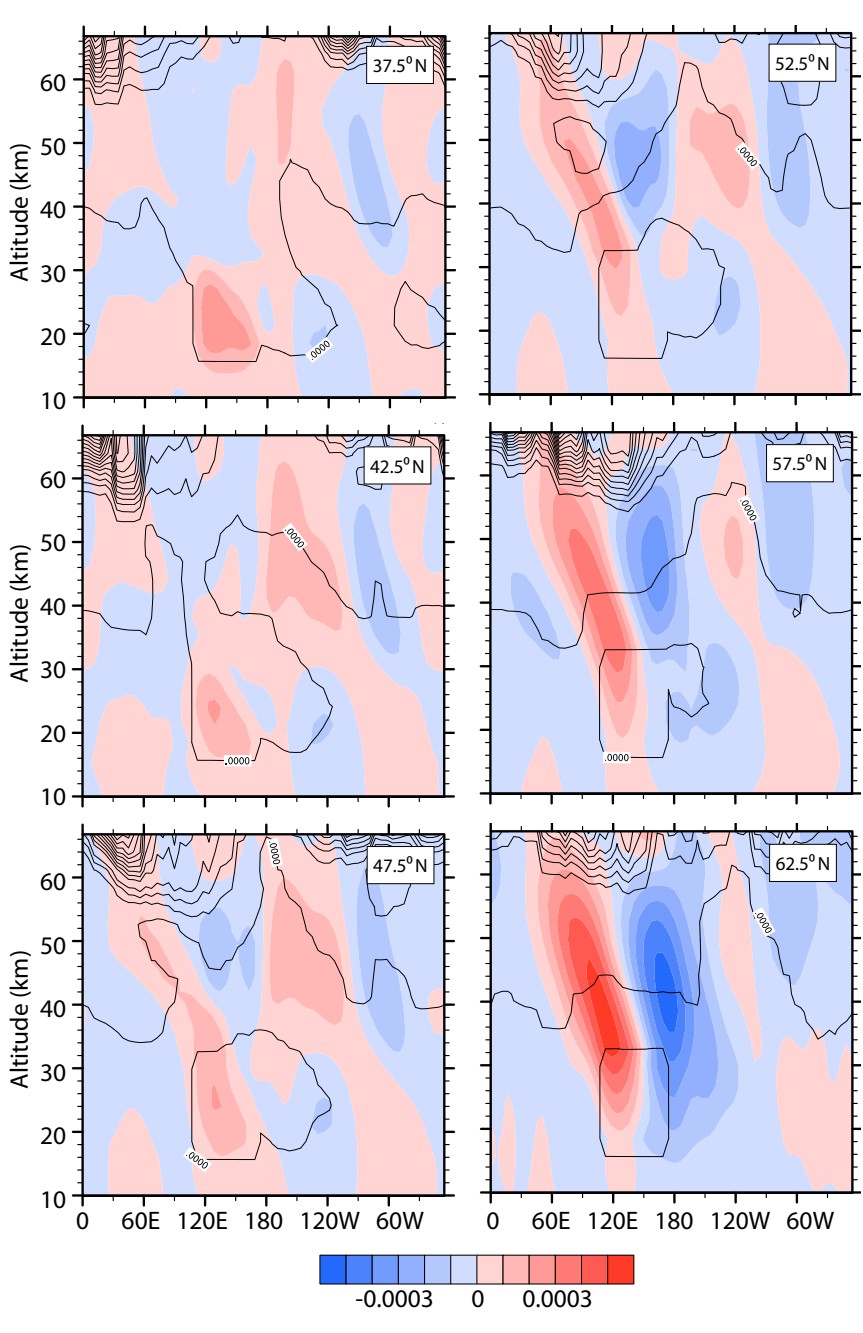

**Fig. 10: Mean January longitudinal cross-sections of Box0.1 simulation residual vertical velocity anomalies [ms⁻¹] at selected latitudes. The contours show gcu to illustrate the position of artificial GWD.**