# Peer review of "Influence of the spatial distribution of gravity wave activity on the middle atmospheric circulation and transport."

_Atmospheric Chemistry and Physics, 2016_

## Referee Comment (RC1) · Anonymous Referee #3 · 10 Aug 2016

The authors explore the impact of zonally localized gravity wave drag on the stratospheric circulation in an idealized model of the middle and upper atmosphere. While the zonal mean residual circulation depends only on the zonal mean wave driving in the downward control limit, the authors show that the zonal mean circulation does in fact depend on the zonal structure of the wave driving. The key is that the zonal structure of the gravity wave drag impacts the resolved wave driving, modifying the propagation of existing planetary waves in addition to generating new waves.

I think there are hints of new and important results in this manuscript, but that the paper requires major revision before it would be acceptable for publication. I hope that the authors interpret the length of my review as a genuine interest in their results, and

desire to help make this paper more effective.

Major Concerns

1) The results of the paper are based on a number of 30 day simulations with the MUA (Middle and Upper Atmosphere) Model. In the real atmosphere, there is substantial natural variability, and results based on a single 30 day snap shot would likely be meaningless in a statistical sense. I suspect that this model does not have much natural variability – otherwise the authors would not be able to conclude much from such short runs – but that is unclear in the current paper, which provides little insight into the background flow and no discussion of the statistics.

To remedy this situation, I recommend first establishing the quality of the model, better characterizing its January climatology (for example, showing the zonal mean wind as a function of pressure and height) along with the variability (for example, the variance of the zonal mean winds). Does this model vary much at all, or it essential steady, seeking only to capture the climatological mean circulation. It would also be good to show the overall impact of the gravity wave drag (GWD) scheme. A panel/overlay showing the zonal mean drag as a function of pressure and latitude might help, too, giving us a better sense the background gravity wave driving.

Then, how sensitive are these quantities to the forcing? At the top of page 4 the authors suggest they force planetary waves 1, 2, and 3 from ERA-I reanalysis at 1000 hPa. Do they mean climatological waves (based on what period)? What would happen if you took the waves from a given year? My concern here is that the authors need to establish that their results are robust, and wouldn't change dramatically if the climatology is altered. Varying the lower boundary would allow them to sample the natural variability of the real world; in his case they would need need to run a number of simulations for each case, and could assess the statistical robustness of their conclusions.

And finally, all figures need to acknowledge the statistics. I don't mean to be the curmudgeon who rants that a result without an error bound isn't a scientific result, but

you do need to either estimate the statistical certainty, or explain that everything that is shown is robust, given the lack of variability in the model.

2) Following up on my first concern, it is unclear to me what these experiments are seeking to represent. Many figures (e.g. 2, 4, etc.) show the 30 day mean, which initially suggested to me that the goal was to demonstrate the steady response to the wave driving (which I presumed had occured over this time scale). But it was not until the Fig. 5 that I realized that the response had clearly not converged over this period!

Gravity waves in the real world tend to be episodic and highly itermittent, so that the short term response is highly relevant. But if the goal is to capture the short term response, then I think the paper needs to focus on this from the start, and establish the appropriate time scale early on. This could be done, for example, by showing a Hovmoller diagram of some key quantities, such as the zonal mean wind at 6 hPa (or another key level) as a function of latitude and time, along with the evolution of the evolution of the key zonal harmonics (as in Fig. 5), but again, plotted as a function of latitude and time. The goal would be to show that the key change(s) occur on a timescale of X days (where X is with hope < 30 days!), establishing that a short 30 day run is sufficient for the study. And then subsequent figures could focus on the key time period(s). I say periods because Fig. 5 hints that there is some oscillitory nature to the response.

I still worry, however, that the short term response may depend a lot on the initial condition as discussed above. For example, in the real world, the propagation and breaking of gravity waves will be very different if the polar vortex is very strong vs. very weak (i.e. after a Sudden Warming). So one ideally would want to sample over different background states to robustly establish the short term response. [I assume the authors are forcing the model with some climatological mean wave forcing, but would it make a difference to use waves from a given year, etc.?]

I do appreciate that the authors have provided information about the time evolution in

supplementary videos, but I feel that the time evolution is vital to the paper, and can't be left in the supplement.

3) I think it would help the paper to organize around key scientific question(s) and results. The discussion/conclusion section was more a discussion of other papers, and left me a bit confused as to what *this* paper was trying to say. In it's current form, the paper comes across as a bit descriptive, e.g. we tried this, and this happened. I appreciate that this is how science often moves forward, but in the conclusions, I urge them to step back and summarize how these simulations do give us new understanding.

I really think there is a lot of potential material here, just that the authors need to better focus the paper. Here are two key areas that could be the main result – just one or would be sufficient – and I don't mean to restrict the authors to these points.

(a) Based on my own interests, I was particularly excited about the zonal mean response to zonally asymmetric wave driving. Given that downward control indicates that the time mean residual circulation depends only on the zonal mean wave driving, one might think that the zonal structure of the gravity wave driving should not matter. But since zonally asymmetric GWD induces a response in resolved waves, the *total* zonal mean wave driving depends very much on the zonal structure of the GWD. To show this, downward control analysis and more discussion of the compensation and interaction between resolved and parameterized wave driving would help.

The authors acknolwedge that nudging might limit the zonal mean response, and so drive compensation by itself. Initially I though the nudging was done to "improve the troposphere", but upon re-reading, I realized it extends to 30 km, fairly deep into the stratosphere! How strong is the nudging in the stratosphere? Can you estimate it's effective amplitude, and compare it to that of the applied gravity wave driving? Downward control can still be applied, but you just need to account for the torque produced by the nudging.

(b) Another important and novel key result could be the impact of localized GWD on

the overall resolved wave structure, following up on the Holton (1984) result that asymmetric GWD generates planetary waves. In this case, I think the time evolution of the flow is much more important. The key would be to establish how fast the resolved flow responds to the gravity wave driving, the dependence on the background state, the linearity of the response, and so forth. These results are in the paper, but I just feel they get lost in the discussion at the end.

Note that result (a) is more about the steady/climatological response, while (b) would be more about the time evolution. Once you know the targeted result, earlier figures could help lead the way.

4) Overall, the presentation of the paper needs to be improved. Small things, such as keeping the names of the simulations uniform and avoid non-standard acronyms (e.g. "gcu, gcv, gt"), and keeping the orientation of the latitudinal axes constant, really do help the reader. I appreciate that the first author is a student, and when I look back at my first papers, I'm embarrassed by the barely perceptible contour lines and tiny font size of the figures. So please take the comments below as suggestions on how the presention could be improved, not as an attempt to be overly critical.

And as will come out in the detailed comments below, I think the paper relies to much on supplementary material. In my opinion, it's okay to have additional figures/movies for the curious reader, but all the key results of the paper should be within the paper.

Other suggestions by page:line number

(And note that it would be good to include page numbers, as the line numbers reset on each page. I hope I have kept things straight myself!)

1:28-34 This first sentence about Holton 1983 is a bit confusing/vague, and then there is a giant leap of 30 years to the present.

1:39 Perhaps you could say "ozone and greenhouse gases" instead of "climate change gases". Also, the references here are for the mesosphere and thermosphere, but not

the stratosphere. For the stratosphere, observed temperature trends have been a bit more puzzling, e.g. Thompson et al. 2012, Nature.

More generally, how does the second paragraph on climate change relate to the results of this paper? If you really want to cover all of climate change in the middle atmosphere, and how well models appear to simulate it, you would need a lot more references. But I this would be taking the paper off track. It might be sufficient to shorten this paragraph and direct the reader to review papers that highlight the significance of the stratosphere (e.g. Kidston et al. Gerber et al.,) and recent analyses of the CMIP5 models, (e.g. Charlton et al. 2013 and Manzini et al. 2014), which assess the "state of the art" when it comes to modeling. I think the goal should be to quickly get across the message that the stratosphere matters, and then zero in on your topic.

2:10 "The BDC is still..."

More generally, it's my understanding that this paragraph is trying to highlight the fact that the "BDC" is a slippery creature to define. It was first discovered based on the distribution of trace gases by Brewer (1949) and Dobson (1956). It is often quantified by the residual circulation (Dunkerton 1978), which can be closely linked with the isentropic circulation. But tracers with geographically varying sources/sinks (such as ozone) are also transported by Rossby waves along isentropic surfaces, a process referred to as isentropic mixing. To understand the movement of water vapor or ozone, you need to account both for the residual mean transport and the isentropic mixing. Plumb (2002) is a good paper to highlight this. However, you can still make a lot of progress with tracer distributions in a 2D context, based on the interplay between the residual circulation and mixing. The three dimensional structure is a new frontier in research, seeking to explain the detailed 3-D structure of temperature and trace gases.

2:22 I would say that the discussion on pre-conditioning is still an active area of research, though "agreement" is building.

3:3 "studies are"

page 3 general comment.

There have been a few other studies that have considered the impact of localize wave torques, and they would be relevant to your discussion.

Shaw, T. A., and W. R. Boos, 2012: The tropospheric response to tropical and subtropical zonally-asymmetric torques: Analytical and idealized numerical model results. J. Atmos. Sci., 69, 214-234.

Naftali Y. Cohen, and William R. Boos, 2016: Modulation of subtropical stratospheric waves by equatorial rainfall, Geophysical Research Letters, 43, 466–471, doi: 10.1002/2015GL067028

In addition, this paper follows up on Cohen et al. 2013 to discuss the mechanism behind compensation in greater detail. I mention it because it discusses the time scale of the response to forcing. In the stratospheric surf zone, they find it's very quick, reaching near equiliberium 5-10 days.

Naftali Y. Cohen, Edwin P. Gerber and Oliver Bühler, 2014: What drives the Brewer-Dobson circulation? Journal of the Atmospheric sciences, 71, 3837–3855, doi: 10.1175/JAS-D-14-0021.1

First paragraph on 2.1. It might help the reader to explain a bit more about MUAM here. I gather that the model includes a troposphere, as the bottom is 1000 hPa, but is the troposphere very unrealistic, given the fairly coarse resolution? Does is it have an active tropospheric circulation with synoptic variability, or is the troposphere passive, and simply there to communicate the surface planetary wave forcing up to the tropopause? Explaining a bit more detail about the nudging might be appropriate here, too. How strong is it above the tropopause?

For context, there are models that just capture the middle atmosphere, e.g. Scott and Polvani (2006), where the lower boundary condition is the geopotential height near the tropopause? Here, the lower boundary to the stratosphere is completely specified, but

it was clear to me how it works in MUAM.

4:6-8 To explain my confusion above, this sentence suggest that "PW and tides" are added. Perhaps the authors mean, "develop", as they are internally generated, right? They are not specified exactly, as implied by "added". [My apologies if this is just a linguistic issue.]

4:7 Does the model spin up to a steady state? Or is it chaotic (like the real atmosphere), that it spins up eddies, etc.. and the initial condition does matter.

4:13 It might be more helpful to report the source stress than the velocity amplitude.

4:20 gcu, gcv, and gt are not intuitive acronyms; it's not even clear what the "c" is supposed to mean.

4:35 Even though the parameters do not change, I believe that the drag can change in response to changes in the resolved flow. It might be good to emphasize this, especially if these changes are not trivial.

4:36 Cohen et al. 2013 suggest that sharp changes in gravity wave forcing are highly likely to be compensated, as the resulting circulation wouldn't be stable otherwise.

5:4 By "not covered by ... the reference run" do the authors simply mean that there is no enhanced gravity wave drag in these longitudes in the reference parameters.

5:5-12 Alexander and Rosenlof 1996 make some useful estimates of the "missing" drag that is likely explained by gravity waves. At 10 hPa, the estimate values around -1 m/s. (In general, I think we do know that net effect of gravity waves, at least in the lower stratosphere, is to decelerate the flow. Palmer et al. 1986, a pioneering study on gravity wave parameterization, added gravity waves to slow down the flow. I was therefore a bit surprised that net effect of gravity waves in Fig. 1 was generally a positive acceleration in the winter hemisphere. What level is shown here?

Section 2.1 general comment: I may have missed it, but it was hard for me to find

the vertical structure of the enhanced gravity wave drag. The horizontal structure is detailed at 4:29, but where do you explain the vertical structure? I assume the net acceleration is constant in height?

5:30-38 This is what I was trying to get out in my comment on page 2 and the BDC. It is not trivial to match the residual circulation, or even the three dimensional residual circulation, to tracer distributions because mixing plays a large role in their transport.

Table 1 I strongly recommend a uniform naming convention for your simulations. Why switch from Box0.1 to Box0.5 to 10Box to SSWBox. The last simulations should be Box10 and Box70 for consistency.

Also, what is "pos" in Box0.1pos supposed to indicate? I guess you mean than additional positive (northerly) wave drag has been added, but it's not clear why this makes it "pos".

6:13 I don't really understand why you call this an SSW. It is true that putting a massive wave drag into the stratosphere kills the vortex, but is this really an SSW? Is it sudden, or does the vortex simply decelerate in response to the massive drag?

6:20-23 I do not understand this sentence.

6:23-24 before discussion positive/negative interference, it might be good to establish that these anomalies are indeed linear.

General comment on Figure 2: I found this figure hard to interpret. It might help to break it down a bit (or at least discuss it more slowly), to first help the reader understand the basic response of the model, and then it's sensitivity to different features.

It would help a lot to include titles above each plot, as I was constantly going up and down from the caption trying to understand what I was looking at. And the contour interval / color scale is changing all over the place. It's okay to use different color scale for the total field vs. anomaly fields, but otherwise, please fix them, so one can more easily compare panels.

This comment on titles over plots and uniform color scale applies to subsequent figures, too.

6:33-7:4 It might be good to refocus the figure on these key results that you want to show.

It seems rather intuitive to me that the local (zonally asymmetric) response to Box gravity wave should be larger than the zonal gravity wave: the local amplitude is much larger when you focus it on a narrow region. Is the zonal mean response that much different? This seems to be a more relevant (and potentially interesting) question. As the zonal mean forcing is the same in both cases, should we expect the zonal mean response to be the same?

7:2-4 This discussion on nudging was a bit disconcerting. That's why I recommend explaining it in more detail to the reader earlier (as noted above).

Does the nudging imply that the zonal uniform response to the gravity wave drag is largely constrained, such that my questions above the zonal mean response above can't really be asked with this model?

7:5-18 This discussion was confusing for me. Are the authors comparing the response to their gravity wave perturbation at 6.25 hPa with the response in to greenhouse gas forcing at 850 hPa in He at al. 2015? If so, this makes little sense. I suggest removing this paragraph entirely, or explaining why this comparison is relevant.

7:17-18 Why would you expect this? And again, how can you compare the response of the mid stratosphere to the near surface?

More generally, is the response linear; i.e. if you increase the forcing by a factor of 10, does the response scale up by a factor of 10? This would be a good thing to establish.

7:19-25 The connection between these results and SSWs is unclear to me. As with the preceding paragraph, I don't think it belongs in the paper.

7:26-33 As noted in my major comments, the time evolution is extremely important. And the fact that this is not a climatological (converged) response makes the comparison with global warming even more tenuous.

7:36 If I am not mistaken, the forcing is exactly 7 times stronger than in the 10box run. Is the anomaly approximately 7 time stronger?

7:34-8:5 The authors need to show the time evolution here if they want to relate this to a SSW. How sudden is the warming? Is it simply a massive gravity wave drag destroying the vortex, or does the resolve circulation play a role in the break down of the vortex. [All this said, I'm not sure how relevant this simulation is to the real world, or to the key conclusions of this paper.]

8:5-7 What is the vertical extent of the gravity wave driving? I presume that it does not extend to 60 km.

8:8-11 I'm willing to accept this is as the best default run, but I was not convinced by the discussion here. If the response is linear, then it's trivial to chose an integration. If it is not linear, then it would be good to give more motivation why this is the best case.

8:25-26 Why is this unexpected. If I understand correctly, the net acceleration of the imposed gravity wave drag is constant in height. But since there is more mass lower in the atmosphere, the effective drag is much larger at the bottom (more precisely, the net force will be proportional to pressure).

In Fig 4, I believe the authors scale the E-P flux divergence as net force, so I'd expect the response to be largest at the bottom – this is simply where perturbation force is largest. (I suspect the response is in part a compensation, as explored by Cohen et al. 2013,14.)

8:27-29 Note that the E-P fluxes are really just a diagnostic. They don't establish causality. Thus it might be better to say "The anomalies are associated with a stronger poleward ...

To establish causality, you would need to explain why the waves propagate more strongly poleward.

9:2-5 As noted above, it is clear that the gravity wave drag caused these changes in wave propagation (you compare with and without gravity wave drag). But it's not clear to me why this is happening?

Is it a compensating response? Or could it be interpreted with the index of refraction, that the deceleration by the gravity wave drag slows the winds, causing the resolved waves to shift and break in new places?

9:6-8 Why does this happen? Why do you need to presume that it creates poleward propagating PWs?

General comment: there's too much discussion of supplementary material. If it's important, please include it in the paper.

9:15-17 "Probably" doesn't sound very scientific. What is the basis for this speculation? [And does the gravity wave deceleration extend below 35 km, or it confined above this level. If you think nudging is active here, then the response of the model is probably questionable.]

9:20 By total harmonic amplitude, do you mean the RMS amplitude? Or the mean square amplitude?

9:18-26 These figures suggest that the time evolution is quite complex, and that the simulations have not converged over 30 days. If the time evolution is important, it should be explored and discussed in more detail. It's not clear to my why, for example, you get a peak response at day 6.

9:27-35 I believe that the inertial gravity waves generally don't have a period of 1 day. The frequency is related to the Coriolis parameter, and so a function of latitude. Gravity wave frequencies is bounded between the Brunt-Vaisala frequency N and the Coriolis parameter f, with interial or near inertial waves coalescing at f = 2 omega sin (lat). At

(for example) 50 N, it's 16 hours, and it will only will only be a day at a single latitude in the subtropics.

More generally, why would you expect the forcing to radiative inertial gravity waves? There's definitely something odd here. If the forcing is causing instability, I'd be quite worried about the ability of the model to resolve it, given the coarse resolution.

10:1-2 I do not understand this argument. Is the response really periodic on longer time scales? It's unclear from just a 30 day snap shot – and the solution doesn't seem to have converged to a periodic oscillation at either latitude.

A Hovmoller diagram would allow you to show the wave amplitude as a function of latitude and time, providing valuable information in the paper that is now in the supplement.

10:23-31 It is hard to follow or understand this discussion without more evidence. What is the evidence of wave reflection? Does it show up in a change in the vertical structure of the waves?

I'm curious how these anomalous waves related to the climatological waves. Is there positive or negative interference (see Fletcher and Kushner 2011)

11:22-27 The time evolution is very troubling, and not sufficiently document in the text.

Note also that the shallow branch of the residual circulation is generally associated with synoptic waves breaking on the top of the subtropical jet. Are there synoptic waves in your model? If you want to explore the residual circulation in more detail, I might suggest considering a downward control analysis. To what extent are the resolved waves amplifying or compensating the anomalies associated with the artificial drag? [You could include the nudging in this analysis: if it's not too strong, it might not overwhelm the response.]

11:28-12:20 I don't think it's appropriate to spend so much of the text discussing the supplement. If this is important, please hovmoller diagrams or other means to distill

[Figure]

into a figure that can be included in the paper.

Fig. 8. It was unclear to me how to relate this figure to the results shown in the paper. Are the authors arguing that there is enhanced subsidence is causing the enhanced ozone column? Is so, please show this.

12:34-13:5 I think it is quite a stretch to compare MUAM vertical velocity anomalies with MIPAS CH4. Why not start by comparing with the vertical velocity in a reanalysis, such as MERRA, which I believe extends pretty high in the stratosphere. Then you could compare the same quantity.

13:34-35 What discrepancy? Do you mean the differences that are only shown in supplementary figures? The "SSW" run is an extremely nonlinear case, so I don't think you should expect it to be similar, and I'm not sure how relevant it is to the real world.

14:4-12 I do not really understand the discussion in this paragraph. What positive feed-back? I suspect gravity waves break in the EA/NP region because they are produced by flow over the Tibetan plateau and instability in the storm track. They break when they reach critical levels or become convective unstable. I don't see how the temperature in the region, however, would cause them to break.

14:33-38 This would be an interesting result, but I don't think the compensation argu-ment is really developed in the paper.

As noted in my major comments, I feel that the conclusions section become a narrative of open questions and interesting results in the field, but is not very much related to the results of the paper. This section needs to be reworked in detail. I think a shorter summary and discussion would make the paper more effective. (Overall I recommend using the conclusions section to review the key results of the paper and explain their broader context.)

Fig 1. What vertical level(s) are shown here?

Fig. 4 Not much happens in the austral (summer) hemisphere here, so you could focus

just on the boreal hemisphere, allowing you more space to show the key changes.

Also, generally people have the north pole on the left.

Fig 7. Might be better to simply plot the stream function, as opposed to arrows.

Fig 9 Units are missing. It would also help to show the same vertical extent in the upper and lower panels.

Fig 10. It would help to show the location of the enhanced gravity wave driving in the figure, for reference.
* * *

---

## Referee Comment (RC2) · Anonymous Referee #1 · 12 Aug 2016

In their ACPD paper *"Influence of the spatial distribution of gravity wave activity on the middle atmospheric circulation and transport"*, Sacha et al. discuss a set of model experiments to analyse the influence of spatial variations of gravity wave activity on various aspects of the middle atmospheric circulation. The study raises some interesting aspects on the impact of localized gravity wave enhancements and their uncertainties, however, there are a number of issues concerning structure, method and presentation of the study. Before publication, please see below for a list of issues as well as technical corrections that should be considered by the authors.

- P1L31: Please specify "and other global datasets"

[Figure]

- P1L38: Specify "changes of the middle atmosphere"

- P2L25: What other phenomena cause abrupt changes of MA circulation? If not relevant, please rephrase.

- P2L32: Explain here (shortly) again why. Mention again PW generation by IGW if relevant.

- P2L39: processes such as e.g. ...? List shortly.

- P4L2: What do you mean with "introduce"? Do you mean prescribe? Nudge?

- P4L8: How long is the spin-up period?

- P4L10: Mean January conditions? Of what period? Or a specific January? In this context, take into account the comments of referee #3

- P4L25-26: Can this abrupt change lead to dynamic instabilities during the transition?

- P4L22-28: Should this GWD modification be understood as rather a change in orographic or non-orographic GWs or as a mixture? Can you estimate that from the observations in Sacha et al. 2015?

- P4L37: Please explain why you are not smoothing the boundaries and if that could have any effect.

- Table 1: Please explain better the systematic behind these experiments. Many values of the table cannot be found in the text. Are these values random guessing (trial and error) or is there a particular science question behind every combination of values?
  Caption: Note that the gcu ...

- Sect. 2.2: It should be made a lot clearer in this section what can be compared here. The model does not seem to calculate interactive chemistry for ozone and methane, and these tracer distributions do not reflect purely dynamical effects (which is mentioned). At this point I do not see how 30 day model simulations (with only January conditions) are supposed to be compared with 30 year annual climatologies of satellite observations. Moreover, are these satellite observations in well enough resolution (temporal and spatial) to hold for comparisons with the effects studied in the model?

- P6L6: Explain why you analyse the 6.25hPa level.

- P6L13: The SSW simulations have not been explained before (only in the table). There should be information in the main text about those.

- P6L21-33: This paragraph should be revised comprehensively. Fig. 2 should be split into two or three figures, in the print-out version, the wind vectors are hardly visible and also the other features are not clear. The meaning of the mentioned results are not clear (particularly line 24-26) and the sentence from line 27-30 should be split to make the points one by one. Also, the word "quite" in line 28 should be removed or specified.

- P7L26-28: This mechanism should be explained better and/or citations included.

- P7L30-32: This should be in the discussion and outlook section, maybe the entire paragraph.

- P8L5: over how much time is that strengthening and shift taking place?

- P8L11-12: This statement should be constrained further in such way that the robustness of this behaviour has not been tested in other vortex situations.

- P8L15: How can I see that this vortex displacement is more rapid?

- In general: At many places, line breaks should be used instead of blank lines everywhere. This would help to divide the respective sections into individual units of meaning.

- Fig 4: You do not discuss Fig. 4d in the text here, instead you mention one "not shown" figure and one figure from the Supplement. You should consider to restructure this. Also, I would appreciate the contoured lines for the box of enhanced GW drag, as in 4f, in all panels. However, why are there 3 lines, is it not always the same box?

- P10L12: This (rather abstract) figure should be introduced with some motivation why you plot this and/or what you expect to learn from plotting this.

- P10L34: Explain why you choose a level slightly above.

- P12L18: What else can it be? And what does that mean for the simulations?

- P13L13-21: This comparison does not seem sensible to me. $CH_4$ is influenced by much more than only vertical velocity (chemistry, advection, diffusion) and thus the comparison does not hold. Also, the patterns you describe in the plots are hardly visible and the motivation for this comparison is not clear to me either. I am not sure if the comparison is crucial for your results anyway, since you do not conclude any vital points here, but if so, the comparison should be made much more carefully.

- Section 4: I think this section should be revised comprehensively. Now, it is some mishmash of discussion, outlook (partly irrelevant like P16L36), conclusions and literature review (partly with only little relation to the results of this study, e.g. P17L8-12). It should be structured more thoroughly around the results of this study and link the findings more clearly to the literature (e.g. P15L26-32: It feels like there lacks a (half) sentence at the end that integrates your results into the

ones from the references mentioned). The second paragraph is a literature review without any clear connection to the results, rather, it raises questions that cannot be answered with this setup; that seems out of place. The second paragraph discusses some insufficiancies of the idealistic modelling approach. This is indeed very important, but it is not made clear, what that means for the conclusions that can be drawn from the model experiments (what can/could still be learned out of the vortex displacement simulation even though it is not reliable?). The connection of your results with the PDO should be discussed more thoroughly because from your model setup (mean January) you cannot compare different PDO phases. Moreover, the PDO had never been mentioned before in the paper.

**There should be a separate and concise conclusions section that lists the main findings of this study** (one of which e.g. an extract of the last paragraph of the paper, this is a very important point).

**Technical corrections**

- Supplement: Please use video formats as in the guidelines for ACP papers (http://www.atmospheric-chemistry-and-physics.net/for_authors/manuscript_preparation.html)

- Supplement: Captions: Use units as suggested by ACP guidelines (e.g. $kg \cdot s^{-2}$)

- Supplement: Please convert the three text files into one with captions on the same pages as figures.

- Supplement: The reference vectors should have max one decimal place and possibly be the same in each panel. As is, the real values are very hard to estimate and to compare.

- P1L21: variability of the Brewer...

[Figure]

- P2L1: ..., yet...

- P2L2: Change "projections" to "simulations" (projections cannot be compared with observations!)

- P2L6: Hence, better ...

- P2L8: ... equations, Dunkerton ...

- P2L21: ... and later on, e.g. ...

- P2L23: ... preconditioning sudden stratospheric warming (SSW ...) events.

- P2L25: SSWs belong ...

- P2L30: replace "come along" with "correlate"

- P2L31: preconditioning SSWs

- P3L6: ... showed that orographic gravity wave induced drag in the lower stratosphere can significantly affect the development of SSWs ...

- P3L10: orographic gravity wave

- P3L10-11: Please restructure to something like: Recently, McL found changes of ...., through artificially modifying ...

- P3L14: ...to reach an energetically more favorable...

- P3L16: In this study, ...

- P3L21: insert comma after brackets

- P3L24: ...the role of the GWD distribution and of the artificial forcing components ...

- P3L31: Finally, we show the differences of the BDC due to the geometry of the GWD modulation and ...

- P3L31: ...4, we...

- P4L10: "for all simulations" instead of "here"

- P4L14: As input for the parameterization scheme, we modified the GW ... field of the potential ...

- P4L26: incorporate GWD modifications.

- P4L32: averaged

- P6L1: Why "subjectively"? This should be objectively and/or explained better.

- P6L5: The header sounds confusing. Suggestion: Atmospheric response to variations in GWD

- P6L35: remove "one may see that"

- P6L36: use "large" (or alike) instead of "significant" unless you conducted the statistical test here.

- P7L5: Therefore, ...

- P7L9: remove "which is chosen for ...our results".

- P7L11: ... 1m/s. This is comparable with the ...

- P7L13: ...level, however, we cannot investigate this because our model has an idealistic troposphere. Still, the location of the GW hotspot in this region can be relevant for the ...
- P7L25: remove "an"

- P7L37: ... in Fig. 2. Therefore, we ...

- P8L3: So you mean right after the spin-up, right? Suggestion: In the SSWbox simulation, immediately after the spin-up period when the GWD is artificially modified, a formation ...

- P8L5-8: Please rephrase: Here, the anomaly is ten times stronger than in ... and hence we can observe ... and the vortex ...

- P8L36: What do you mean with "Putting together", please rephrase.

- P9L4: ... simulation, the ..

- P9L5: Include (Fig. 4c) after anomalies

- P9L8: ... (Fig 4e), also pronounced anomalous equatorward propagation can be observed.

- P9L10: Move (Fig. 4b) to end of sentence.

- P9L12: Therefore, we ...

- P9L12-16: The sentence is unclear and very long, should be split and restructured.

- P9L22: Here, we ...

- P9L37: ... amplitude, the peak ...

- P10L7-10: The sentence is unclear and very long, should be split and restructured.

- P10L20: We can find the maximal amplitude anomalies at ...

- P10L26-29: The sentence is unclear and very long, should be split and restructured. (and "one may see" should be replaced)

- P10L37: ... and the wave-1 maxima are descending to altitudes below 30 km over time.

- P11L7: Replace biggest with largest.

- P11L7: ...stratosphere, but these results are not directly comparable. The E-P flux ...

- P11L13: remove "very"

- P11L20: south of the enhancement region

- P11L22: If we take into account the total values (..), we can ...

- P11L31: In the anomalies (...), the ...

- P11L36: Remove "approximately above the GWD enhancement area"

- P12L2: This, together with the previously unexpected fact of a weak upper BDC branch response to the artificial GWD can rather be explained by the effect of the monthly averaging than by the locality of the residual circulation response to the artificial GWD (see Animation 4 in the Supplement). At particular time steps, the magnitude of the anomalies is comparable regardless of the BDC branch, but for the deep BDC branch the anomalies oscillate.
  And still I don't think I get the meaning of this paragraph. It should be rephrased with better explanations of the points that you want to make here.

- P12L12: the "other" one

- P12L16: ... after the start the first signs of anomalies evolve.

- P14L14: At first, we discuss ...

- P15L3: If we consider that the intermittent ...

- P15L33: ... during the "positive phase of the(?)" PDO ...

- P16L1-4: Very long and confusing sentence. Please rephrase.

- Fig. 10: The only nubers in the contours is .0000. Why so many digits? Add other numbers, this way it is impossible to see how much the variation is. Add another number to the colour bar to make counting easier, in this context, think about using $x \cdot 10^{-4}$ Caption: Add the unit to gcu.
* * *

---

## Short Comment (SC1) · 12 Aug 2016

On behalf of all co-authors I would like to thank the referee for his comments, constructive feedback and the positive attitude towards improving the paper.

I would also like to inform the referee about our intention to upload our response rather gradually in parts- to give him the chance for possible interactive reaction. The simulations are quite time-demanding and depending on the number of additional simulations needed, we might not be able to react to the complete set of comments before the end of the interactive discussion.

Finally, I would like to point out, that some of the specific comments related to the

comment 4 are outdated, since they relate to the original version of the manuscript not to the current version in ACPD. The manuscript underwent numerous changes during the revision process for publication in ACPD. For reference, I am enclosing the current version with marked changes from the original version.

Currently,together with the co-authors, we are discussing the comments 1) and 2). We acknowledge the referee for pointing out the "artificiality" of the 30 day mean in our results. We were aware of this, and therefore we always discussed the related time evolution and presented supplementary figures and animations to illustrate it. As the referee was afraid of - in this experimental set-up, we cannot claim that the key changes occur on a time scale <30days (See the attached Fig.1-Hovmoeller plot of amplitude of the first harmonic from FT of zonal wind at approx. 6hPa).In Fig. 1 one can also see the propagation of oscillations to the SH, which was erroneously assigned to inertia GWs, as was pointed out by the referee. Fig. 2 presents Hovmoeller plot of zonal mean zonal wind at 6hPa from a reference simulation.

I would like to thank you very much again and the first part of response will be uploaded soon.

Petr Šácha.

Please also note the supplement to this comment:
http://www.atmos-chem-phys-discuss.net/acp-2016-548/acp-2016-548-SC1-supplement.pdf
* * *
[Figure]

[Figure]

Fig. 1.

[Figure]

Fig. 2.

**Supplement:**

[revised manuscript text omitted]

---

## Referee Comment (RC3) · Anonymous Referee #2 · 17 Aug 2016

The paper by Sacha et al. investigates the impact of a locallized GW forcing on the circulation of the middle atmosphere. This is a highly interesting topic which merrits publication in ACP. I support the comment of Referee#3 that for fully appreciating the results, it is necessary to assess the variability of the model and the influence of this on the results. There is one point, in addition, I would like to comment on.

Both the authors and Referee#3 emphasize the intermittency of gravity waves. The quoted intermittency investigations focus on the high variability considering single waves / individual observations. This intermittency may be used for instance to develop / improve GW parametrization schemes (de la Camera et al., JGR, 2014). The situation is different, however, if we consider regional averages. Regional averages for

regions with prominent mountain wave forcing also yield highly intermittent GW variances and GWMF, with variations of more than an order of magnitude from day to day (e.g. Eckermann and Preusse, 1999, Jiang et al., JGR, 2002, Schroeder et al., GRL, 2009). The situation is different, for instance, for subtropical convective gravity waves (i.e. summer subtropics). Considering single wave events, there is also large intermittency between events. GWMF and also other wave parameters (phase speed, wavelengths) are highly variable. Considering a larger region as in the current paper, the average behavior however does vary much less (e.g. Schroeder et al., GRL, 2009). For the wintertime forcing discussed here, shear would be a likely source (e.g. Leena et al., JASTP, 2012; Pramitha et al., ACP, 2015; Atmos. Res. 2016). Unfortunately, we have for this forcing a lack of sufficiently frequent remote sensing observations (i.e. insufficient temporal resolution), in order to quantify the temporal variability, but it may be argued that also the winter time regional average would not lead to strong pulses (i.e. day-to-day variations). Thus, assuming a constant forcing after the onset of some general meteorological condition, seems a plausible assumption and therefore focussing on the average response after a few days a plausible approach.

Concerning the comment of Referee#3 about highlighting the novel findings of the paper. I find the discussion section interesting and relevant. How about adding a very short summary section which just highlights a few findings?
* * *

---

## Short Comment (SC2) · 19 Aug 2016

On behalf of all co-authors I would like to thank the referee for his numerous and detailed comments, questions and suggestions.

Also in connection to the comments from Referees #2 and #3, we can summarize the response to the major comments accordingly:

1)We will revise the model description part together with the description of experiments and variability.

2)The tracer related discussion will be removed and the figures will be moved to the supplement - only to highlight the motivation for research of the EA/NP region.

[Figure]

3)The Discussion and conclusions section will be revisited to give primarily a clear summary of the presented results.

I would like to thank you very much again. Hopefully, a point to point author response will be uploaded soon.

Petr Šácha.

---

## Short Comment (SC3) · 19 Aug 2016

On behalf of all co-authors I would like to thank the Referee #2 for giving his opinion on the comments raised by other referees, especially, for the contribution to the discussion on the intermittency of GWs.

I would like to ask the referee to follow the upcoming author responses to Ref#1 and Ref#3, where your points are going to emerge. In the end of the discussion, of course, we will summarize a response to your comments as well.

With many thanks, Petr Šácha.

---

## Author Comment (AC1) · 7 Sep 2016

**ACP review:**
**Ref#3** comments in italic:

This response is concerned with a general response to some important topics raised by the referee. A point to point response including a detailed list of changes will come later.

At first, we would like to thank the referee for pointing out the paper Cohen et al. (2014). This paper will greatly help us to generalize our results in future work.

However, our analysis was motivated with the possible effect of the EA/NP GW hotspot and only during the work we realized that it can have also general consequences in contrasting the effect of zonally symmetric and asymmetric torgues.

It is tempting to repeat the methodology given by Cohen et al. (2014), but we would rather leave it for future work, while keeping the main focus of our paper on the analysis of the local effects of the GWD injection. We will refer to Cohen et al. (2014), and also to Shaw and Boos (2012), who, for example, found the same pattern in the vertical velocity response (with downwelling to the northeast and upwelling to the southwest from the zonally asymmetric torgue), although in the troposphere.

Below is our response to selected referee's comments that are given in italic.

*1A) The results of the paper are based on a number of 30 day simulations with the MUA (Middle and Upper Atmosphere) Model. In the real atmosphere, there is substantial natural variability, and results based on a single 30 day snap shot would likely be meaningless in a statistical sense. I suspect that this model does not have much natural variability – otherwise the authors would not be able to conclude much from such short runs – but that is unclear in the current paper, which provides little insight into the background flow and no discussion of the statistics. To remedy this situation, I recommend first establishing the quality of the model, better characterizing its January climatology (for example, showing the zonal mean wind as a function of pressure and height) along with the variability (for*
*example, the variance of the zonal mean winds). Does this model vary much at all, or it essential steady, seeking only to capture the climatological mean circulation. It would also be good to show the overall impact of the gravity wave drag (GWD) scheme. A panel/overlay showing the zonal mean drag as a function of pressure and latitude might help, too, giving us a better sense the background gravity wave driving.*

This is right, because a steady state is nudged at the lower boundary, MUAM does not contain day-to-day variability and captures the climatological mean circulation. The only variable component are the solar tides. We agree that the quality of the MUAM′s January climatology needs to be demonstrated. We will add additional figures in the section 2.1.

*Then, how sensitive are these quantities to the forcing? At the top of page 4 the authors suggest they force planetary waves 1, 2, and 3 from ERA-I reanalysis at 1000 hPa. Do they mean climatological waves (based on what period)? What would happen*

*if you took the waves from a given year? My concern here is that the authors need to establish that their results are robust, and wouldn't change dramatically if the climatology is altered. Varying the lower boundary would allow them to sample the natural variability of the real world; in his case they would need need to run a number of simulations for each case, and could assess the statistical robustness of their conclusions.*

We agree with this comment. The overall response will be different for different background conditions. But there is a question, how reasonable it is to compare the responses of different background states to the same GWD. Naturally, each different background state means different GW sourcing and propagation conditions that should result to a different GWD. With our artificial enhancement approach we are not able to reflect this. Instead, in our simulations we have chosen a different approach. For mean January conditions we inject GWD settings stemming from our estimation based on observations (and from uncertainties especially in its meridional component) and from the mean climatology in the EA/NP region. In our paper we wanted to shed the light on the acting mechanisms and patterns of local response (e.g. build up of a positive geopotential anomaly upstream from the GWD, obstacle analogy..) that we assume will remain valid also for sligthly different conditions.

Actually, we have also made simulations with slightly different background climatology and the results were almost similar (absolute agreement in the pattern, only the magnitude was slightly different in some places). We would not like to show the figures in the revised manuscript, because we think that it would have little additional value, since in our manuscript we are not exactly concerned with the precise magnitude of the response.

*1B) And finally, all figures need to acknowledge the statistics. I don't mean to be the curmudgeon who rants that a result without an error bound isn't a scientific result, but you do need to either estimate the statistical certainty, or explain that everything that is shown is robust, given the lack of variability in the model.*

We absolutely agree. In a revised manuscript all of the mean fields will be overlaid with stippled areas with the significance exceeding the 5% level (p- value estimated using the t-test). From Fig. R1 it can be seen that e.g. for the mean EPflux difference between Box0.1and Ref simulation the area of most important differences, which were discussed in the text, is signifficant.

[Figure]

CONTOUR FROM .01 TO 1 BY .2475

0.01   0.05   0.25   0.5   1

**Fig. R1:Plot of p-value computed by t-test. On the x-axis is latitude and on the y-axis is a log-pressure height in km.**

*2Aa) Following up on my first concern, it is unclear to me what these experiments are seeking to represent. Many figures (e.g. 2, 4, etc.) show the 30 day mean, which initially suggested to me that the goal was to demonstrate the steady response to the wave driving (which I presumed had occured over this time scale). But it was not until the Fig. 5 that I realized that the response had clearly not converged over this period!*

The response to each artificial GWD in each simulation is unsteady to some extend in the sense that it constantly radiates the oscillations (their nature is discussed later in this response), but this effect can be canceled by averaging. On longer scales we can see some background evolution in response to the GWD, supposingly connected with PW creation. Previously in the manuscript, we felt that the 30day average is somewhat artificial in this respect and therefore we tried to always supplement the averaged results with information of time evolution (Fig. 5), animation (Fig. 3, Fig. 4, Fig. 6, Fig.7) or discussion on time evolution within the 30 day mean.

Following the comments made by the referee, in the revised manuscript we will at first establish from Hovmöller diagrams the time interval, which can be considered

as steady (from ca 6 days to the end of simulation, see Fig. R2). This will allow us to compute mean responses, but only for simulations with -0.5 and -10 m/s/day artificial GW induced zonal acceleration. For -70 m/s/day the simulations do not reach a steady state during the simulation. The responses in those "SSW simulations" were always presented as snapshots in selected time steps or as animations only. In the revised version, in addition, we will present time evolution of zonal mean zonal wind to prove the SSW like nature of the vortex events.

**Hovmoeller plot of zonal mean zonal wind at 6hPa**

[Figure]

**Fig. R2: Hovmöller plot of zonal mean zonal wind from 10box simulation.**

*2Ab)Gravity waves in the real world tend to be episodic and highly itermittent, so that the short term response is highly relevant. But if the goal is to capture the short term response, then I think the paper needs to focus on this from the start, and establish the appropriate time scale early on. This could be done, for example, by showing a Hovmoller diagram of some key quantities, such as the zonal mean wind at 6 hPa (or another key level) as a function of latitude and time, along with the evolution of the evolution of the key zonal harmonics (as in Fig. 5), but again, plotted as a function of latitude and time. The goal would be to show that the key change(s) occur on a timescale of X days (where X is with hope < 30 days!), establishing that a short 30 day run is sufficient for the study. And then subsequent*
*figures could focus on the key time period(s). I say periods because Fig. 5 hints that there is some oscillitory nature to the response.*

As written above, we will present Hovmöller diagrams at the start of the Results section. Averaging will start ca 6 days after GWD injection, when the large-scale respond is building up (note the agreement with the time scale of the transient response build up in Cohen et al. (2014), where it is related to the life time of PW breaking).

*2Ba)I still worry, however, that the short term response may depend a lot on the initial condition as discussed above. For example, in the real world, the propagation and breaking of gravity waves will be very different if the polar vortex is very strong vs. very weak (i.e. after a Sudden Warming). So one ideally would want to sample over different background states to robustly establish the short term response. [I assume the authors are forcing the model with some climatological mean wave forcing, but would it make a difference to use waves from a given year, etc.?]*

This is true, but the problem is that our GWD injection (value and ratio) is constant, not taking into account the background. With a little exaggeration we can say that for each time step in each ensemble member we would need another GWD enhancement values and only then we can sample for a robust response. We think that the presented results stemming from injection of constant GWD (estimated in accordance with the mean background conditions) are good enough to analyze the nature of the response as well as the discrepancy between effects of localized and zonally symmetric forcing.

*2Bb) I do appreciate that the authors have provided information about the time evolution in supplementary videos, but I feel that the time evolution is vital to the paper, and can't be left in the supplement.*

At this point, we have to acknowledge the referee's suggestion to use Hovmöller plots. By presenting lat-time plots of the zonal mean variance and of harmonic amplitudes we can replace Figure 5 and Animation 2. Also, by presenting the time evolution of zonal mean zonal wind for SSW simulations we can give a more detailed information about the polar vortex evolution, not relying on the animations only.

*3) I think it would help the paper to organize around key scientific question(s) and results. The discussion/conclusion section was more a discussion of other papers, and left me a bit confused as to what \*this\* paper was trying to say. In it's current form, the paper comes across as a bit descriptive, e.g. we tried this, and this happened. I appreciate that this is how science often moves forward, but in the conclusions, I urge them to step back and summarize how these simulations do give us new understanding.*
*I really think there is a lot of potential material here, just that the authors need to better focus the paper. Here are two key areas that could be the main result – just one or would be sufficient – and I don't mean to restrict the authors to these points.*

We would like to keep the structure unchanged. This means division into polar vortex, PW and residual circulation response sections. But, thanks to the referee's comments, we will now make it clear in the paper, where, how and why the steady state and the transient response is analysed.

The discussion section will be revised and a clear summary of results will be given in the start (to satisfy Ref#1 and #2 as well). Otherwise, it is probably true that the paper can be marked as descriptive - in the sense that we are more concerned with the qualitative than the quantitative analysis of our results.

*(3a1) Based on my own interests, I was particularly excited about the zonal mean response to zonally asymmetric wave driving. Given that downward control indicates that the time mean residual circulation depends only on the zonal mean wave driving, one might think that the zonal structure of the gravity wave driving should not matter. But since zonally asymmetric GWD induces a response in resolved waves, the \*total\* zonal mean wave driving depends very much on the zonal structure of the GWD. To show this, downward control analysis and more discussion of the compensation and interaction between resolved and parameterized wave driving would help.*

As written at the beginning of the response, we consider a more generalized experimnet at a later stage (we have a three-years grant for this research). At this moment, we have to leave it for a future work (near future), also for following reasons:
To precisely imitate the methodology of Cohen et al. (2014) it would mean to produce a set of new and very long simulations (with reduced nudging, probably) and we are not able to manage it in the period reserved for the manuscript revision. Also, for a precise estimation of the compensation ratio we would have to modify our method of GWD enhancement (e.g. to enhance the GWD by multiplying the GW parametrization output in a box). Because by prescribing a constant GWD we are missing an important and very quick response of GWs to the modified PW field. We will discuss it as a wekness in our qualitative approach, but this would probably severely harm any quantitative results.

For your interest, we show in Fig. R3 the degree of compensation computed for the response to a localized -0.5 m/s/day artificial GW induced zonal acceleration and -0.1 m/s/day meridional acceleration (Box0.1 simulation). The ratio shows almost a perfect compensation in the southern part of the artificial GWD area, but a strong amplification in the northern part - in accordance with the E-P flux divergence anomaly (Fig. 4) in the manuscript. The ratio (Fig. R3, bottom) of degrees of compensation between Box0.1 (C=0.01) and Zon0.1 (C=0.07) simulation shows (the sum in the tittle) that the degree of compensation is in a sum across the domain stronger, because there is less amplification in the Zon0.1 simulation. In the ratio (Fig. R3, bottom) emerge also relatively large differences in the distribution of compensation in the regions with almost zero compensation (small number/even smaller).

[Figure]

[Figure]

**Fig. R3: Top: Lat-lev distribution of the mean degree of compensation for Box0.1 simulation. Bottom: Lat-lev distribution of the degree of compensation for Box0.1/ degree of compensation for Zon0.1 simulation.**

*(3a2) The authors acknolwedge that nudging might limit the zonal mean response, and so drive compensation by itself. Initially I though the nudging was done to "improve the troposphere", but upon re-reading, I realized it extends to 30 km, fairly deep into the stratosphere! How strong is the nudging in the stratosphere?*
*Can you estimate it's effective amplitude, and compare it to that of the applied gravity wave driving? Down- ward control can still be applied, but you just need to account for the torque produced by the nudging.*

We shall take this into account. Generally, the extent of nudging is a main reason why we analyze the response at the altitude around 35km and not at some level where GWD is enhanced.

*(3b) Another important and novel key result could be the impact of localized GWD on he overall resolved wave structure, following up on the Holton (1984) result that asymmetric GWD generates planetary waves. In this case, I think the time evolution of the flow is much more important. The key would be to establish how fast the resolved flow responds to the gravity wave driving, the dependence on the background state, the linearity of the response, and so forth. These results are in the paper, but I just feel they get lost in the discussion at the end.*
*Note that result (a) is more about the steady/climatological response, while (b) would be more about the time evolution. Once you know the targeted result, earlier figures could help lead the way.*

In the revised manuscript, using Hovmöller diagrams, we are now able to better establish the time scale of the response. But, a detailed analysis of the dependence on a background state is impossible with the concept of artificial injection, as discussed under comment (2Ba).

Regarding the linearity, we do not have enough simulations to perform a robust linearity test (e.g. it is sure that there will be some bifurcation value of GWD). But, from a simple comparison of a geopotential response to -0.5m/s/day and -10m/s/day localized GWD (Fig. R4), we can see that, globally, the response cannot be regarded as linear (the ratio is variable in space).

Locally, we found near linear response between these two simulations in our results -e.g. the strength of the induced positive geopotential anomaly upstream from the artificial GWD area. But this comparison can be misleading, since both simulations (10box and Box0.1) have the same value of meridional GWD component. This means that the drag force has different orientation between these two simulations!

With certainty we can say that the response to the strongest injection of -70 m/s/day in our SSW simulations is fully nonlinear, as it contains creation of new pressure structures etc.. In the revised manuscript we will always take a great care when mentioning terms like positive or destructive interferrence .

[Figure]

**Figure R4: First row: Mean geopotential height difference between Box0.1 and Ref simulation on the left and 10Box and Ref simulation on the right. Second row: Corresponding standard deviation. Third row: Ratio between Box0.1 and 10 Box geopotential mean difference on the left and ratio of standard deviations on the right. Box0.1 simulation is performed with local enhancement of artificial gravity wave induced zonal acceleration by -0.5m/s/day and 10box with -10 m/s/day. Both have the same local enhancement of artificial gravity wave induced meridional acceleration (-0.1m/s/day).**

*4) Overall, the presentation of the paper needs to be improved. Small things, such as keeping the names of the simulations uniform and avoid non-standard acronyms (e.g. "gcu, gcv, gt"), and keeping the orientation of the latitudinal axes constant, really do help the reader. I appreciate that the first author is a student, and when I look back at my first papers, I'm embarrassed by the barely perceptible contour lines and tiny font size of the figures. So please take the comments below as suggestions on how the presention could be improved, not as an attempt to be overly critical.*
*And as will come out in the detailed comments below, I think the paper relies to much on supplementary material. In my opinion, it's okay to have additional figures/movies for the curious reader, but all the key results of the paper should be within the paper.*

We will follow the specific comments from all three referees to improve the presentation. Regarding the role of supplementary material, in the revised manuscript its importance is diminished, as some information on time evolution is now available from Hovmöller plots. Also the residual circulation part is restructured - plots showing tracer distributions are moved into the Supplement for motivation to study EA/NP only and this makes additional room for Figs. S1 and S2 (or their modification) directly in the text.

*Specific comment: 9:27-35 I believe that the inertial gravity waves generally don't have a period of 1 day. The frequency is related to the Coriolis parameter, and so a function of latitude. Gravity wave frequencies is bounded between the Brunt- Vaisala frequency N and the Coriolis parameter f, with interial or near inertial waves coalescing at f = 2 omega sin (lat). At (for example) 50 N, it's 16 hours, and it will only will only be a day at a single latitude in the subtropics.*
*More generally, why would you expect the forcing to radiative inertial gravity waves? There's definitely something odd here. If the forcing is causing instability, I'd be quite worried about the ability of the model to resolve it, given the coarse resolution.*

We would like to thank the referee for noticing this. We have made a thorough analysis of this oscillating anomalies, coming to the conclusion that the reason is most likely a non-linear interaction between anomalously generated inertia GWs and solar tides (see e.g. Waltersheid, 1981). We have three supporting arguments for such a hypothesis: **1)** the oscillations are present in the SH and NH except polar region (January conditions - no tides at NH polar region). **2)** From NH midlatitudes to SH midlatitudes the oscillations have prevailing period around 24h, but going towards SH polar region the dominant period decreases between 12 and 8 hours. This is consistent with the fact that only inertia GWs with periods lower than $f$ can propagate towards the pole. **3)** We can observe a decrease of longer anamoulous wave modes (lower than 10 cycles/Earth perimeter) when going to the SH polar regions

More generally, we expect the forcing to constantly perturb the predominantly zonal flow and so causing particle displacements. As a result, broad spectrum of waves can radiate from the region in dependence on what kind of waves are allowed by the background conditions. We considered especially the inertia GWs, because their modes can be long enough to be resolved by the model, while propagating sufficiently quickly (Fig. R5) and predominantly horizontally to be accountable for the oscillatory pattern.

**References:**

*Cohen, N. Y., E. P. Gerber, and O. Bühler, 2013: Compensation between resolved and unresolved wave driving in the strato- sphere: Implications for downward control. J. Atmos. Sci., 70, 3780–3798, doi:10.1175/JAS-D-12-0346.1.*
*Naftali Y. Cohen, Edwin P. Gerber and Oliver Bühler, 2014: What drives the Brewer-Dobson circulation? Journal of the Atmospheric sciences, 71, 3837– 3855, doi: 10.1175/JAS-D-14-0021.1*

*Shaw, T. A., and W. R. Boos, 2012: The tropospheric response to tropical and subtropical zonally-asymmetric torques: Analytical and idealized numerical model results. J. Atmos. Sci., 69, 214-234.*

*Walterscheid, R. L. "Inertio-gravity wave induced accelerations of mean flow having an imposed periodic component: Implications for tidal observations in the meteor region." Journal of Geophysical Research: Oceans 86.C10 (1981): 9698-9706.*

---

## Author Comment (AC2) · 21 Oct 2016

**ACP review:**
This is a point to point response to the **Ref#3** comments.
Referee comments are in italic.

At first, we would like to thank the referee for all of his comments, ideas and suggestions. The manuscript has been greatly improved thanks to the time he devoted to our manuscript.

***Major Concerns:***
*1) The results of the paper are based on a number of 30 day simulations with the MUA (Middle and Upper Atmosphere) Model. In the real atmosphere, there is substantial natural variability, and results based on a single 30 day snap shot would likely be meaningless in a statistical sense. I suspect that this model does not have much natural variability – otherwise the authors would not be able to conclude much from such short runs – but that is unclear in the current paper, which provides little insight into the background flow and no discussion of the statistics.*
*To remedy this situation, I recommend first establishing the quality of the model, better characterizing its January climatology (for example, showing the zonal mean wind as a function of pressure and height) along with the variability (for example, the variance of the zonal mean winds). Does this model vary much at all, or it essential steady, seeking only to capture the climatological mean circulation. It would also be good to show the overall impact of the gravity wave drag (GWD) scheme. A panel/overlay showing the zonal mean drag as a function of pressure and latitude might help, too, giving us a better sense the background gravity wave driving.*

Answer: We absolutely agree and therefore we made major changes in the manuscript:
In section 2.1, the model, its variability and the procedure of creation of the sensitivity simulations are desribed in more detail. Information about the reference simulation is given in Fig.1. In Figure 3 and at the start of the Results section we now establish the time scale of the response and statistics is now included everywhere in the manuscript, where mean anomalies or differences are dealt with. Standard deviations of the zonal means in the reference simulation in the 30 days are small as described in the text (P4L10-15).

*Then, how sensitive are these quantities to the forcing? At the top of page 4 the authors suggest they force planetary waves 1, 2, and 3 from ERA-I reanalysis at 1000 hPa. Do they mean climatological waves (based on what period)? What would happen if you took the waves from a given year? My concern here is that the authors need to estab- lish that their results are robust, and wouldn't change dramatically if the climatology is altered. Varying the lower boundary would allow them to sample the natural variability of the real world; in his case they would need need to run a number of simulations for each case, and could assess the statistical robustness of their conclusions.*

Answer: The waves are extracted from decadal mean ERAI reanalysis and are stationary only (P3L35). The words "decadal monthly mean" have been included

(P3L30) to point that out. The model reaches a steady state after the spin-up (Fig. 3A).

More general answer from the AC1:

We agree with this comment. The overall response will be different for different background conditions. But there is a question, how reasonable it is to compare the responses of different background states to the same GWD. Naturally, each different background state means different GW sourcing and propagation conditions that should result to a different GWD. With our artificial enhancement approach we are not able to reflect this. Instead, in our simulations we have chosen a different approach. For mean January conditions we inject GWD settings stemming from our estimation based on observations (and from uncertainties especially in its meridional component) and from the mean climatology in the EA/NP region. In our paper we wanted to shed the light on the acting mechanisms and patterns of local response (e.g. build up of a positive geopotential anomaly upstream from the GWD, obstacle analogy..) that we assume will remain valid also for sligthly different conditions.

Actually, we have also made simulations with slightly different background climatology and the results were almost similar (absolute agreement in the pattern, only the magnitude was slightly different in some places). We would not like to show the figures in the revised manuscript, because we think that it would have little additional value, since in our manuscript we are not exactly concerned with the precise magnitude of the response.

*And finally, all figures need to acknowledge the statistics. I don't mean to be the curmudgeon who rants that a result without an error bound isn't a scientific result, but you do need to either estimate the statistical certainty, or explain that everything that is shown is robust, given the lack of variability in the model.*

Answer:We absolutely agree and thank the referee for pointing out this. In a revised manuscript all of the mean (from day 7 to 30) differences and anomalies are overlaid with stippled areas of significance exceeding the 5% level (p- value estimated using the t-test). Except the new Fig. 9, where the standard deviation is dotted.

*2) Following up on my first concern, it is unclear to me what these experiments are seeking to represent. Many figures (e.g. 2, 4, etc.) show the 30 day mean, which initially suggested to me that the goal was to demonstrate the steady response to the wave driving (which I presumed had occured over this time scale). But it was not until the Fig. 5 that I realized that the response had clearly not converged over this period!*
*Gravity waves in the real world tend to be episodic and highly itermittent, so that the short term response is highly relevant. But if the goal is to capture the short term response, then I think the paper needs to focus on this from the start, and establish the appropriate time scale early on. This could be done, for example, by showing a Hovmoller diagram of some key quantities, such as the zonal mean wind at 6 hPa (or another key level) as a function of latitude and time, along with the evolution of the evolution of the key zonal harmonics (as in Fig. 5), but again, plotted as a function of latitude and time. The goal would be to show that the key change(s) occur on a timescale of X days (where X is with hope < 30 days!),*

*establishing that a short 30 day run is sufficient for the study. And then subsequent figures could focus on the key time period(s). I say periods because Fig. 5 hints that there is some oscillitory nature to the response.*

Answer: The paper has been significantly improved in terms of structure, clarity and description of results. We now also make it absolutely clear what our simulations seek to represent (P6L30-38) - mean response to a monthly mean GWD distribution (as is usual from satellite observations). Following the comments made by the referee, we use Hovmöller diagrams to establish the time interval, which can be considered as quasi-steady (from ca 7 days to the end of simulation, see Fig. 3). This allows us to compute mean responses, but only for simulations with -0.5 and -10 m/s/day artificial GW induced zonal acceleration. For -70 m/s/day the simulations do not reach a steady state during the simulation. The responses in those "SSW simulations" are always presented as snapshots in selected time steps (Figs. 6, 8) or as animations (Animation 1a, b, 2 and 4). In the revised version, in addition, we present time evolution of zonal mean zonal wind to prove the SSW like nature of the vortex events (Fig. 4).

*I still worry, however, that the short term response may depend a lot on the initial condition as discussed above. For example, in the real world, the propagation and breaking of gravity waves will be very different if the polar vortex is very strong vs. very weak (i.e. after a Sudden Warming). So one ideally would want to sample over different background states to robustly establish the short term response. [I assume the authors are forcing the model with some climatological mean wave forcing, but would it make a difference to use waves from a given year, etc.?]*

Answer from the AC1: This is true, but the problem is that our GWD injection (value and ratio) is constant, not taking into account the background. With a little exaggeration we can say that for each time step in each ensemble member we would need another GWD enhancement values and only then we can sample for a robust response. We think that the presented results stemming from injection of constant GWD (estimated in accordance with the mean background conditions) are good enough to analyze the nature of the response as well as the discrepancy between effects of localized and zonally symmetric forcing.

*I do appreciate that the authors have provided information about the time evolution in supplementary videos, but I feel that the time evolution is vital to the paper, and can't be left in the supplement.*

Thanks to the referee's suggestion we use Hovmöller plots where the information on time evolution was important (Fig. 3, 4, 10). The animations in the supplement are now rather for interested readers. The paper is consistent without them and the volume of text discussing them has been minimalized. But we strongly encourage everyone to take a look at the animations, since some features are much more clear from the animation.

*3) I think it would help the paper to organize around key scientific question(s) and results. The discussion/conclusion section was more a discussion of other papers, and left me a bit confused as to what *this* paper was trying to say. In it's current*

*form, the paper comes across as a bit descriptive, e.g. we tried this, and this happened. I appreciate that this is how science often moves forward, but in the conclusions, I urge them to step back and summarize how these simulations do give us new understanding.*

Answer: The organization of the paper has not been changed. This means division into polar vortex, PW and residual circulation response sections. But thanks to the comments from all referees, the entire Results section has been improved. Discussion section has been restructured to be more focused and the summary of results is given P14L28-P15L24.

*I really think there is a lot of potential material here, just that the authors need to better focus the paper. Here are two key areas that could be the main result – just one or would be sufficient – and I don't mean to restrict the authors to these points.*
*(a) Based on my own interests, I was particularly excited about the zonal mean response to zonally asymmetric wave driving. Given that downward control indicates that the time mean residual circulation depends only on the zonal mean wave driving, one might think that the zonal structure of the gravity wave driving should not matter. But since zonally asymmetric GWD induces a response in resolved waves, the \*total\* zonal mean wave driving depends very much on the zonal structure of the GWD. To show this, downward control analysis and more discussion of the compensation and interaction between resolved and parameterized wave driving would help.*
*The authors acknolwedge that nudging might limit the zonal mean response, and so drive compensation by itself. Initially I though the nudging was done to "improve the troposphere", but upon re-reading, I realized it extends to 30 km, fairly deep into the stratosphere! How strong is the nudging in the stratosphere?*
*Can you estimate it's effective amplitude, and compare it to that of the applied gravity wave driving? Down- ward control can still be applied, but you just need to account for the torque produced by the nudging.*

Answer: For a detailed response to the first paragraph, please see the AC1 - In short, this is a highly interesting topic and we want to perform such an analysis in near future.
Regarding the nudging, it is dependent on the strength of the zonal mean response. It acts on zonal mean temperatures in MUAM and the magnitude ranges from lesser than 1K/day for the reference simulation to almost 2K/day for the SSWbox simulation (See Fig. 1c in the revised manuscript). An explanation of the role of nudging in MUAM (P3L37-P4L5) and a figure of the strength of nudging for the SSWbox have been added (Fig. 1c). To minimalize the effect on our results we analyze (except Figs. 7 and 8, 11 and 12) our results two layers above the nudging upper boundary.

*(b) Another important and novel key result could be the impact of localized GWD on he overall resolved wave structure, following up on the Holton (1984) result that asymmetric GWD generates planetary waves. In this case, I think the time evolution of the flow is much more important. The key would be to establish how fast the resolved flow responds to the gravity wave driving, the dependence on the*

*background state, the linearity of the response, and so forth. These results are in the paper, but I just feel they get lost in the discussion at the end.*
*Note that result (a) is more about the steady/climatological response, while (b) would be more about the time evolution. Once you know the targeted result, earlier figures could help lead the way.*

Answer: In the revised manuscript the section 3.2 has been rewritten to focus more on the PW generation by GWD. Also, time evolution is taken into account (new Fig. 10). But as stated in AC1, a detailed analysis of the dependence on a background state is impossible with the concept of artificial injection we are using.

*4)Overall, the presentation of the paper needs to be improved. Small things, such as keeping the names of the simulations uniform and avoid non-standard acronyms (e.g. "gcu, gcv, gt"), and keeping the orientation of the latitudinal axes constant, really do help the reader. I appreciate that the first author is a student, and when I look back at my first papers, I'm embarrassed by the barely perceptible contour lines and tiny font size of the figures. So please take the comments below as suggestions on how the presention could be improved, not as an attempt to be overly critical.*
*And as will come out in the detailed comments below, I think the paper relies to much on supplementary material. In my opinion, it's okay to have additional figures/movies for the curious reader, but all the key results of the paper should be within the paper.*

Answer: The text in all sections underwent major revisions. The names of simulations were uniform already in the previous version - according the naming convention (except the SSW and specific simulations that are better characterized by shorter names): "*gcu + distribution + gcv + sign of gcv*", where gcu = -0.5 m/s/day is not stated in the name. This is now made clear at P4L35.
The acronyms are unchanged (*"gcu, gcv, gt"*) and are properly defined in the text (P4L26-27), because we are not aware about any standard acronym for those GWD components.

**Other suggestions by page:line number:**

*1:28-34 This first sentence about Holton 1983 is a bit confusing/vague, and then there is a giant leap of 30 years to the present.*
Answer: Deleted.

*1:39 Perhaps you could say "ozone and greenhouse gases" instead of "climate change gases". Also, the references here are for the mesosphere and thermosphere, but not the stratosphere. For the stratosphere, observed temperature trends have been a bit more puzzling, e.g. Thompson et al. 2012, Nature.*
Answer: Deleted.

*More generally, how does the second paragraph on climate change relate to the results of this paper? If you really want to cover all of climate change in the middle atmosphere, and how well models appear to simulate it, you would need a lot more*

*references. But I this would be taking the paper off track. It might be sufficient to shorten this paragraph and direct the reader to review papers that highlight the significance of the stratosphere (e.g. Kidston et al. Gerber et al.,) and recent analyses of the CMIP5 models, (e.g. Charlton et al. 2013 and Manzini et al. 2014), which assess the "state of the art" when it comes to modeling. I think the goal should be to quickly get across the message that the stratosphere matters, and then zero in on your topic.*

Answer: The paragraph has been shortened and the reference Manzini et al. 2014, added. P1L34

*2:10 "The BDC is still..."*
*More generally, it's my understanding that this paragraph is trying to highlight the fact that the "BDC" is a slippery creature to define. It was first discovered based on the distribution of trace gases by Brewer (1949) and Dobson (1956). It is often quanti- fied by the residual circulation (Dunkerton 1978), which can be closely linked with the isentropic circulation. But tracers with geographically varying sources/sinks (such as ozone) are also transported by Rossby waves along isentropic surfaces, a process re- ferred to as isentropic mixing. To understand the movement of water vapor or ozone, you need to account both for the residual mean transport and the isentropic mixing. Plumb (2002) is a good paper to highlight this. However, you can still make a lot of progress with tracer distributions in a 2D context, based on the interplay between the residual circulation and mixing. The three dimensional structure is a new frontier in re- search, seeking to explain the detailed 3-D structure of temperature and trace gases.*

Answer:The paragraph about BDC has been rewritten based on the referee comments (P1L38-P2L7).

*2:22 I would say that the discussion on pre-conditioning is still an active area of re- search, though "agreement" is building.*

Answer: The text was changed to (P2L11-12): There is a building agreement in the literature on the role of wave activity in preconditioning sudden stratospheric warming

*page 3 general comment.*
*There have been a few other studies that have considered the impact of localize wave torques, and they would be relevant to your discussion.*
*Shaw, T. A., and W. R. Boos, 2012: The tropospheric response to tropical and subtrop- ical zonally-asymmetric torques: Analytical and idealized numerical model results. J. Atmos. Sci., 69, 214-234.*
*Naftali Y. Cohen, and William R. Boos, 2016: Modulation of subtropical strato- spheric waves by equatorial rainfall, Geophysical Research Letters, 43, 466–471, doi: 10.1002/2015GL067028*
*In addition, this paper follows up on Cohen et al. 2013 to discuss the mechanism behind compensation in greater detail. I mention it because it discusses the time scale of the response to forcing. In the stratospheric surf zone, they find it's very quick, reaching near equiliberium 5-10 days.*
*Naftali Y. Cohen, Edwin P. Gerber and Oliver Bühler, 2014: What drives the Brewer-Dobson circulation? Journal of the Atmospheric sciences, 71, 3837–3855, doi: 10.1175/JAS-D-14-0021.1*

Answer: Thank you for bringing those papers into our attention. Studies Shaw and Boos (2014)-P14L10- and Cohen et al. (2014) -P3L4, P6L37, P9L37, P16L7.

*First paragraph on 2.1. It might help the reader to explain a bit more about MUAM here. I gather that the model includes a troposphere, as the bottom is 1000 hPa, but is the troposphere very unrealistic, given the fairly coarse resolution? Does is it have an active tropospheric circulation with synoptic variability, or is the troposphere passive, and simply there to communicate the surface planetary wave forcing up to the tropopause? Explaining a bit more detail about the nudging might be appropriate here, too. How strong is it above the tropopause?*
*For context, there are models that just capture the middle atmosphere, e.g. Scott and Polvani (2006), where the lower boundary condition is the geopotential height near the tropopause? Here, the lower boundary to the stratosphere is completely specified, but it was clear to me how it works in MUAM. 4:6-8 To explain my confusion above, this sentence suggest that "PW and tides" are added. Perhaps the authors mean, "develop", as they are internally generated, right? They are not specified exactly, as implied by "added". [My apologies if this is just a linguistic issue.]*
*4:7 Does the model spin up to a steady state? Or is it chaotic (like the real atmosphere), that it spins up eddies, etc.. and the initial condition does matter.*
*4:13 It might be more helpful to report the source stress than the velocity amplitude.*

Answer: As we only use monthly mean reanalysis data the troposphere is not meant to be a representation of a synoptical state but rather a monthly mean climatology. This has been pointed out, now (P3L30). Furthermore, we included a sentence making clear why the troposphere is necessary and why it needs to be corrected (P3L33).The word "added" is replaced by "generated"(P4L7). An additional figure with zonal mean winds, temperature and GWD has been included (Fig. 1), standard deviations of the zonal mean within these 30 days are small as described in the text (P4L10-P4L15). Nudging is described at P3L37-P4L5.

*4:35 Even though the parameters do not change, I believe that the drag can change in response to changes in the resolved flow. It might be good to emphasize this, especially if these changes are not trivial.*
Answer: We agree that it should be like this, but it is not the case in our simulations, as stated at P5L4-5. In our sensitivity simulations the GW parameterization scheme is switched off and we use the GW parameterization output from the reference simulation.

*4:36 Cohen et al. 2013 suggest that sharp changes in gravity wave forcing are highly likely to be compensated, as the resulting circulation wouldn't be stable otherwise.*
Answer: This is now referenced at P5L9-11.

*5:4 By "not covered by … the reference run" do the authors simply mean that there is no enhanced gravity wave drag in these longitudes in the reference parameters.*

Answer: The sentence has been rewritten to: There are no exceptional GWD values in the reference simulation in this region. P4L39.

*5:5-12 Alexander and Rosenlof 1996 make some useful estimates of the "missing" drag that is likely explained by gravity waves. At 10 hPa, the estimate values around -1 m/s. (In general, I think we do know that net effect of gravity waves, at least in the lower stratosphere, is to decelerate the flow. Palmer et al. 1986, a pioneering study on gravity wave parameterization, added gravity waves to slow down the flow. I was therefore a bit surprised that net effect of gravity waves in Fig. 1 was generally a positive acceleration in the winter hemisphere. What level is shown here?*

Answer: In Fig.2 (old Fig.1), GWD is plotted at approximately 14hPa (11th model level). We must confess that, at the start of our analysis, we were not satisfied with the performance of the MUAM GW parameterization output in the stratosphere and therefore we have chosen to modify it artificially.

*Section 2.1 general comment: I may have missed it, but it was hard for me to find the vertical structure of the enhanced gravity wave drag. The horizontal structure is detailed at 4:29, but where do you explain the vertical structure? I assume the net acceleration is constant in height?*

Answer: The vertical structure is now illustrated in Figs. 7,8, 11 and 12. The artificially added GWD is constant with height, but the total GW acceleration is not constant with height, because the artificial GWD is added to the reference field (very small background value). We included zonal mean latitude-height plots for GWD components from the reference simulation in Fig. 1d, e and f.

*5:30-38 This is what I was trying to get out in my comment on page 2 and the BDC. It is not trivial to match the residual circulation, or even the three dimensional residual circulation, to tracer distributions because mixing plays a large role in their transport.*

Answer: We deleted the part relating the residual circulation to the tracer distributions. Plots of tracer distributions are now given in the Supplement only (Fig. 1S, 2S) to give a motivation for research in the EA/NP region.

*Table 1 I strongly recommend a uniform naming convention for your simulations. Why switch from Box0.1 to Box0.5 to 10Box to SSWBox. The last simulations should be Box10 and Box70 for consistency.*

*Also, what is "pos" in Box0.1pos supposed to indicate? I guess you mean than additional positive (northerly) wave drag has been added, but it's not clear why this makes it "pos".*

Answer: We think that the names of simulations were uniform already in the previous version - according the naming convention (except for the SSW simulation and other specific simulations that are better characterized by shorter names): "*gcu + distribution + gcv + sign of gcv*", where gcu = -0.5 m/s/day is not stated in the name. This is now made clear at P4L35.

*6:13 I don't really understand why you call this an SSW. It is true that putting a massive wave drag into the stratosphere kills the vortex, but is this really an SSW? Is it sudden, or does the vortex simply decelerate in response to the massive drag?*

Answer: We have added Hovmöller diagrams to illustrate the time evolution of those observed vortex events (Fig. 4). Of course, the dynamics is visible most easily from Animation 1a and 1b in the supplement.

*6:20-23 I do not understand this sentence.*
*6:23-24 before discussion positive/negative interference, it might be good to establish that these anomalies are indeed linear.*
*General comment on Figure 2: I found this figure hard to interpret. It might help to break it down a bit (or at least discuss it more slowly), to first help the reader understand the basic response of the model, and then it's sensitivity to different features.*
*It would help a lot to include titles above each plot, as I was constantly going up and down from the caption trying to understand what I was looking at. And the contour interval / color scale is changing all over the place. It's okay to use different color scale for the total field vs. anomaly fields, but otherwise, please fix them, so one can more easily compare panels.*
Answer: The description of results shown in Fig. 5 (former Fig. 2) has been significantly improved and clarified. In Fig. 5, titles are now included.

*6:33-7:4 It might be good to refocus the figure on these key results that you want to show.*
*It seems rather intuitive to me that the local (zonally asymmetric) response to Box gravity wave should be larger than the zonal gravity wave: the local amplitude is much larger when you focus it on a narrow region. Is the zonal mean response that much different? This seems to be a more relevant (and potentially interesting) question. As the zonal mean forcing is the same in both cases, should we expect the zonal mean response to be the same?*
Answer: Titles in Fig. 5 now inlude also the sum of the geopotential response over the whole domain showing that the zonal mean response is indeed that different.

*7:2-4 This discussion on nudging was a bit disconcerting. That's why I recommend explaining it in more detail to the reader earlier (as noted above).*
*Does the nudging imply that the zonal uniform response to the gravity wave drag is largely constrained, such that my questions above the zonal mean response above can't really be asked with this model?*
Answer: In Fig. 1c and in the text at P3L37-P4L4 we are discussing the effect of nudging and its possible role. It acts to lower the zonal mean response, but it is not to that degree to affect the results of our analysis.

*7:5-18 This discussion was confusing for me. Are the authors comparing the response to their gravity wave perturbation at 6.25 hPa with the response in to greenhouse gas forcing at 850 hPa in He at al. 2015? If so, this makes little sense. I suggest removing this paragraph entirely, or explaining why this comparison is relevant.*
*7:17-18 Why would you expect this? And again, how can you compare the response of the mid stratosphere to the near surface?*
Answer: This paragraph has been removed.

*More generally, is the response linear; i.e. if you increase the forcing by a factor of 10, does the response scale up by a factor of 10? This would be a good thing to establish.*

Answer: Phrases implying the need of linearity were rephrased in the text and two sentences have been added (P8L14-17) to tackle the issue of linearity of the response. Please see AC1 for a more general response with a simple linearity analysis.

*7:19-25 The connection between these results and SSWs is unclear to me. As with the preceding paragraph, I don't think it belongs in the paper.*

*7:26-33 As noted in my major comments, the time evolution is extremely important. And the fact that this is not a climatological (converged) response makes the comparison with global warming even more tenuous.*

Answer: Both paragraphs have been removed.

*7:36 If I am not mistaken, the forcing is exactly 7 times stronger than in the 10box run. Is the anomaly approximately 7 time stronger?*

Answer: We didn't check this as it makes a little sense to compare these two runs (SSWbox and 10box). With certainty we can say that the response to the strongest injection of -70 m/s/day in our SSW simulations is fully nonlinear, as it contains creation of new pressure structures etc.

*7:34-8:5 The authors need to show the time evolution here if they want to relate this to a SSW. How sudden is the warming? Is it simply a massive gravity wave drag destroying the vortex, or does the resolve circulation play a role in the break down of the vortex. [All this said, I'm not sure how relevant this simulation is to the real world, or to the key conclusions of this paper.]*

Answer: The time evolution is shown in Fig. 4 in the revised manuscript. We didn't analyze the SSWs in detail, because this would be worth a single paper (leaving it for near future), but from Fig. 8 it seems clear that anomalous PWs play a leading role.

*8:5-7 What is the vertical extent of the gravity wave driving? I presume that it does not extend to 60 km.*

Answer: The artificial GWD extends from approx. 20 to 30km.

*8:8-11 I'm willing to accept this is as the best default run, but I was not convinced by the discussion here. If the response is linear, then it's trivial to chose an integration. If it is not linear, then it would be good to give more motivation why this is the best case.*

Answer: We are now giving a detailed argumentation why the Box0.1 simulation is chosen for majority of analyses at P5L20-30.

*8:25-26 Why is this unexpected. If I understand correctly, the net acceleration of the imposed gravity wave drag is constant in height. But since there is more mass lower in the atmosphere, the effective drag is much larger at the bottom (more precisely, the net force will be proportional to pressure).*

*In Fig 4, I believe the authors scale the E-P flux divergence as net force, so I'd expect the response to be largest at the bottom – this is simply where perturbation force is*

*largest. (I suspect the response is in part a compensation, as explored by Cohen et al. 2013,14.)*

Answer: This has been explained by one of the referees already during the review process to ACPD and it is not present in the ACPD version.

*8:27-29 Note that the E-P fluxes are really just a diagnostic. They don't establish causality. Thus it might be better to say "The anomalies are associated with a stronger poleward ...*

*To establish causality, you would need to explain why the waves propagate more strongly poleward.*

Answer:We are using now "is associated with", where the causality is not clear. However, the section 3.2 has been significantly rewritten and now we are giving more evidence for generation of PWs by the GWD region.

*9:2-5 As noted above, it is clear that the gravity wave drag caused these changes in wave propagation (you compare with and without gravity wave drag). But it's not clear to me why this is happening?*

*Is it a compensating response? Or could it be interpreted with the index of refraction, that the deceleration by the gravity wave drag slows the winds, causing the resolved waves to shift and break in new places?*

*9:6-8 Why does this happen? Why do you need to presume that it creates poleward propagating PWs?*

Answer: We are now identifying two acting mechanisms that create anomalies in the PW activity in the section 3.2: index of refraction and anomalous PW generation (absent for ring simulations).

We expect the box to create a large spectrum of waves and their modes, because it is an unbalanced force causing displacements in a balanced and predominantly zonal flow. This is now discussed at P10L9-14 .

*General comment: there's too much discussion of supplementary material. If it's important, please include it in the paper.*

Answer: The role od supplementary material has been diminished by the revisions we made.

*9:15-17 "Probably" doesn't sound very scientific. What is the basis for this speculation? [And does the gravity wave deceleration extend below 35 km, or it confined above this level. If you think nudging is active here, then the response of the model is probably questionable.]*

*9:20 By total harmonic amplitude, do you mean the RMS amplitude? Or the mean square amplitude?*

*9:18-26 These figures suggest that the time evolution is quite complex, and that the simulations have not converged over 30 days. If the time evolution is important, it should be explored and discussed in more detail. It's not clear to my why, for example, you get a peak response at day 6.*

Answer: This paragraph and the corresponding figure have been removed.

*9:27-35 I believe that the inertial gravity waves generally don't have a period of 1 day. The frequency is related to the Coriolis parameter, and so a function of latitude. Gravity wave frequencies is bounded between the Brunt-Vaisala frequency*

*N and the Coriolis parameter f, with interial or near inertial waves coalescing at f = 2 omega sin (lat). At (for example) 50 N, it's 16 hours, and it will only will only be a day at a single latitude in the subtropics.*

Answer: Thank you very much for pointing out this. For a detailed response please see the AC1. It is most likely a product of nonlinear interaction of inertia GWs and tides- P12L3-6.

*More generally, why would you expect the forcing to radiative inertial gravity waves? There's definitely something odd here. If the forcing is causing instability, I'd be quite worried about the ability of the model to resolve it, given the coarse resolution.*

Answer: As written above, we expect the GWD box to create displacements (both horizontal and vertical) and then it is up to the background which waves will be supported to propagate. Also inertia GWs long enough to be resolved are anomalously generated and can propagate even to the SH (revealed by power spectral differences of anomalous modal amplitudes between different latitude bands, not shown).

*10:1-2 I do not understand this argument. Is the response really periodic on longer time scales? It's unclear from just a 30 day snap shot – and the solution doesn't seem to have converged to a periodic oscillation at either latitude.*

*A Hovmoller diagram would allow you to show the wave amplitude as a function of latitude and time, providing valuable information in the paper that is now in the supple- ment.*

*10:23-31 It is hard to follow or understand this discussion without more evidence. What is the evidence of wave reflection? Does it show up in a change in the vertical structure of the waves?*

*I'm curious how these anomalous waves related to the climatological waves. Is there positive or negative interference (see Fletcher and Kushner 2011)*

Answer: This subsection has been removed.

*11:22-27 The time evolution is very troubling, and not sufficiently document in the text.*

*Note also that the shallow branch of the residual circulation is generally associated with synoptic waves breaking on the top of the subtropical jet. Are there synoptic waves in your model? If you want to explore the residual circulation in more detail, I might sug- gest considering a downward control analysis. To what extent are the resolved waves amplifying or compensating the anomalies associated with the artificial drag? [You could include the nudging in this analysis: if it's not too strong, it might not overwhelm the response.]*

*11:28-12:20 I don't think it's appropriate to spend so much of the text discussing the supplement. If this is important, please hovmoller diagrams or other means to distill into a figure that can be included in the paper.*

Answer: The entire Section 3 has been revised not to rely on the supplementary animations. We prefer to present the information on residual circulation using vectors (vres, wres) colored by the strangth of the mass flux, because then certain features nicely emerge (lee wave pattern etc.).

*Fig. 8. It was unclear to me how to relate this figure to the results shown in the paper. Are the authors arguing that there is enhanced subsidence is causing the enhanced ozone column? Is so, please show this.*

*12:34-13:5 I think it is quite a stretch to compare MUAM vertical velocity anomalies with MIPAS CH4. Why not start by comparing with the vertical velocity in a reanalysis, such as MERRA, which I believe extends pretty high in the stratosphere. Then you could compare the same quantity.*

Answer: This subsection has been removed and the figures moved to the supplement to give motivation for the EA/NP research only.

*13:34-35 What discrepancy? Do you mean the differences that are only shown in supplementary figures? The "SSW" run is an extremely nonlinear case, so I don't think you should expect it to be similar, and I'm not sure how relevant it is to the real world.*

*14:4-12 I do not really understand the discussion in this paragraph. What positive feed- back? I suspect gravity waves break in the EA/NP region because they are produced by flow over the Tibetan plateau and instability in the storm track. They break when they reach critical levels or become convective unstable. I don't see how the temperature in the region, however, would cause them to break.*

*14:33-38 This would be an interesting result, but I don't think the compensation argu- ment is really developed in the paper.*

*As noted in my major comments, I feel that the conclusions section become a narrative of open questions and interesting results in the field, but is not very much related to the results of the paper. This section needs to be reworked in detail. I think a shorter summary and discussion would make the paper more effective. (Overall I recommend using the conclusions section to review the key results of the paper and explain their broader context.)*

Answer: We absolutely agree and the Discussion and conclusions section has been restructured, rewritten and a summary of results is now given at P14L30-P15L24.

*Comments on figures:* We have newly generated the figures according to the comments.

---

## Author Comment (AC3) · 21 Oct 2016

**ACP review:**
This is a point to point response to the **Ref#1** comments.
Referee comments are in italic.

We would like to thank the referee for his/her numerous comments, suggestions and technical corrections that made the paper so much better. We really appreciate the enormous amount of time you spent with our manuscript.

*P1L31: Please specify "and other global datasets"*
*• P1L38: Specify "changes of the middle atmosphere"*
*• P2L25: What other phenomena cause abrupt changes of MA circulation? If not relevant, please rephrase.*
*• P2L32: Explain here (shortly) again why. Mention again PW generation by IGW if relevant.*
*• P2L39: processes such as e.g. ...? List shortly.*
Answer: The introduction underwent minor changes based on you and Ref#3 comments.

*P4L2: What do you mean with "introduce"? Do you mean prescribe? Nudge?*
"prescribe" is the right word, it is changed (P3L29)

*P4L8: How long is the spin-up period?*
330 model days, has been added in the text (P3L36)

*P4L10: Mean January conditions? Of what period? Or a specific January? In this context, take into account the comments of referee #3*
Decadal mean January means are used and this is now mentioned in the text (P4L9).

*P4L25-26: Can this abrupt change lead to dynamic instabilities during the transition?*
Answer: Yes. The build up of the response is to see in Hovm. diagrams (Figs. 3, 4 and 9).

*P4L22-28: Should this GWD modification be understood as rather a change in orographic or non-orographic GWs or as a mixture? Can you estimate that from the observations in Sacha et al. 2015?*
Answer: Based on Šácha et al. 2015 and on analysis of CMAM-sd GW parametrization output (not shown) we argue that the majority of GWs in the stratosphere in the EA/NP region in January are of orographic origin (P5L24).

*P4L37: Please explain why you are not smoothing the boundaries and if that could have any effect.*
Answer: To simulate the sudden and localized GW breaking effect and also to mimic the EA/NP hotspot given the coarse resolution of the model. It is very likely that the sharp boundaries will determine some patterns of the response (e.g. the lee wave pattern in Fig. 10b)-P16L2-4. In the Box0.1 simulation the boundaries are not as sharp due to the background GWD from the reference simulation.

*Table 1: Please explain better the systematic behind these experiments. Many values of the table cannot be found in the text. Are these values random guessing (trial and error) or is there a particular science question behind every combination of values?*

Answer: We are describing the choices of the GWD components P5L12-30. To summ up, we are varying the GWD along the rough observational constraints (or estimates) for the GWD above the EA/NP region.

*Sect. 2.2: It should be made a lot clearer in this section what can be compared here. The model does not seem to calculate interactive chemistry for ozone and methane, and these tracer distributions do not reflect purely dynamical effects (which is mentioned). At this point I do not see how 30 day model simulations (with only January conditions) are supposed to be compared with 30 year annual climatologies of satellite observations. Moreover, are these satellite observations in well enough resolution (temporal and spatial) to hold for comparisons with the effects studied in the model?*

Answer: The comparison has been removed and the tracer related information is now given in the supplement only for motivation of research in the EA/NP region.

*P6L6: Explain why you analyse the 6.25hPa level.*

Answer: It is the second level above the artificial GWD and nudging upper boundary. The first level above can be influenced by some interface effects. We have added a clarification of this choice at P6L15-18.

*P6L13: The SSW simulations have not been explained before (only in the table). There should be information in the main text about those.*

*P6L21-33: This paragraph should be revised comprehensively. Fig. 2 should be split into two or three figures, in the print-out version, the wind vectors are hardly visible and also the other features are not clear. The meaning of the mentioned results are not clear (particularly line 24-26) and the sentence from line 27-30 should be split to make the points one by one. Also, the word "quite" in line 28 should be removed or specified.*

*P7L26-28: This mechanism should be explained better and/or citations included.*

*P7L30-32: This should be in the discussion and outlook section, maybe the entire paragraph.*

Answer: The SSW simulations are now introduced in Fig.4 and related discussion. Wind vectors in Fig. 5 (former Fig.2) have been enlarged and the subsection related to Fig. 5 has been rewritten.

*P8L5: over how much time is that strengthening and shift taking place?*

*P8L11-12: This statement should be constrained further in such way that the robustness of this behaviour has not been tested in other vortex situations.*

*P8L15: How can I see that this vortex displacement is more rapid?*

Answer: Fig. 4 has been added to give additional information about SSWs and the description of results is now more precise and clear P8L24-P9L12.

*In general: At many places, line breaks should be used instead of blank lines everywhere. This would help to divide the respective sections into individual units of meaning.*
Answer: We are using the ACP template where blank lines seem to be the only way to divide paragraphs. But we agree that this would be helpfull.

*Fig 4: You do not discuss Fig. 4d in the text here, instead you mention one "not shown" figure and one figure from the Supplement. You should consider to restructure this. Also, I would appreciate the contoured lines for the box of enhanced GW drag, as in 4f, in all panels. However, why are there 3 lines, is it not always the same box?*
Answer: Fig. 7 (former Fig.4) as well as other figures has been newly created and the choice of subplots now fits better to the direction of the text.

*P10L12: This (rather abstract) figure should be introduced with some motivation why you plot this and/or what you expect to learn from plotting this.*
Answer: The motivation for Fig. 9 is at P11L15-18 and its results are newly described.

*P12L18: What else can it be? And what does that mean for the simulations?*
Answer: After the comment from Ref#3 we have made a much deeper analysis with the outcome that this pattern is most likely caused by nonlinear interaction of inertia GWs with tides (P12L5).

*P13L13-21: This comparison does not seem sensible to me. CH4 is influenced by much more than only vertical velocity (chemistry, advection, diffusion) and thus the comparison does not hold. Also, the patterns you describe in the plots are hardly visible and the motivation for this comparison is not clear to me either. I am not sure if the comparison is crucial for your results anyway, since you do not conclude any vital points here, but if so, the comparison should be made much more carefully.*
Answer: This subsection has been deleted.

*Section 4: I think this section should be revised comprehensively. Now, it is some mishmash of discussion, outlook (partly irrelevant like P16L36), conclusions and literature review (partly with only little relation to the results of this study, e.g. P17L8-12). It should be structured more thoroughly around the results of this study and link the findings more clearly to the literature (e.g. P15L26-32: It feels like there lacks a (half) sentence at the end that integrates your results into the ones from the references mentioned). The second paragraph is a literature re- view without any clear connection to the results, rather, it raises questions that cannot be answered with this setup; that seems out of place. The second para- graph discusses some insufficiancies of the idealistic modelling approach. This is indeed very important, but it is not made clear, what that means for the con- clusions that can be drawn from the model experiments (what can/could still be learned out of the vortex displacement simulation even though it is not reliable?). The connection of your results with the PDO should be discussed more thoroughly because from your model setup (mean January) you cannot compare different PDO phases. Moreover, the PDO had never been mentioned before in the paper.*

*There should be a separate and concise conclusions section that lists the main findings of this study (one of which e.g. an extract of the last paragraph of the paper, this is a very important point).*

Answer: We absolutely agree and the Discussion and conclusions section has been restructured, rewritten and a summary of results is now given at P14L30-P15L24.

All of the following technical corrections have been implemented and the animations are now created according to ACP guidelines.

---

## Author Comment (AC4) · 21 Oct 2016

**ACP review:**
This is a point to point response to the **Ref#2** comments.
Referee comments are in italic.

We would like to thank the referee for his detailed insight into the intermittency of GWs and the relevance of our GWD enhancement.

*I support the comment of Referee#3 that for fully appreciating the results, it is necessary to assess the variability of the model and the influence of this on the results.*
Answer: The variability of the model is now better described and all of the mean plots now come with an estimate of statistical significance. Please see the response to Ref#3.

*Both the authors and Referee#3 emphasize the intermittency of gravity waves. The quoted intermittency investigations focus on the high variability considering single waves / individual observations. This intermittency may be used for instance to develop / improve GW parametrization schemes (de la Camera et al., JGR, 2014). The situation is different, however, if we consider regional averages. Regional averages for regions with prominent mountain wave forcing also yield highly intermittent GW vari- ances and GWMF, with variations of more than an order of magnitude from day to day (e.g. Eckermann and Preusse, 1999, Jiang et al., JGR, 2002, Schroeder et al., GRL, 2009). The situation is different, for instance, for subtropical convective gravity waves (i.e. summer subtropics). Considering single wave events, there is also large intermittency between events. GWMF and also other wave parameters (phase speed, wavelengths) are highly variable. Considering a larger region as in the current paper, the average behavior however does vary much less (e.g. Schroeder et al., GRL, 2009). For the wintertime forcing discussed here, shear would be a likely source (e.g. Leena et al., JASTP, 2012; Pramitha et al., ACP, 2015; Atmos. Res. 2016). Unfortunately, we have for this forcing a lack of sufficiently frequent remote sensing observations (i.e. in- sufficient temporal resolution), in order to quantify the temporal variability, but it may be argued that also the winter time regional average would not lead to strong pulses (i.e. day-to-day variations). Thus, assuming a constant forcing after the onset of some gen- eral meteorological condition, seems a plausible assumption and therefore focussing on the average response after a few days a plausible approach.*
Answer: Based on your comment we are discussing this at P15L33-35 and the reference Schroeder et al., 2009 is added.

---

## Author Response (AR2)

Author response:

Dear Prof. Dameris,

Thank you and all of the referees very much for such a great review process that led to significant improvements of the manuscript.

We implemented all of the new comments from all referees, except the issue 4) raised by the Ref. #1 - that the Fig. 5 is too small. We have printed it in its current version (after enlargement of Fig. 5 after the interactive discussion) and we find it readable. We want to keep all of the subfigures in Fig.5 and so it is almost impossible to enlarge it further.

Also the issue 3) raised by the Ref. #1 has been adopted only partially, because we didn't delete the paragraph about supporting tracer data and the consequent figures in the Supplement. We softened even more our description of those figures - to be illustrative only, but still we would like to present them to attract researchers to analysis of the peculiar behaviour of the transport in the EA/NP region (and possibly the role of the Aleutian High, or Himalays, which is not stated explicitly in our paper).

On the other side, we adapted the comment to change the title, because we agree that the paper is no longer significantly related with tracer distributions. We think that the word dynamics describes the overall influence of GW distribution even better than the word circulation.

Ref # 1 and 2 also suggested that our discussion can be relevant for orographic GW parameterizations as well - this is a very valuable comment and we are now stating this in the text with two references (Ern et al. 2016 and Yamashita 2013) added.

Other comments were of technical nature and were adopted. You may find the revised version with marked up changes below.

[revised manuscript text omitted]